# A systematic review and meta-analysis of transdiagnostic cognitive behavioural therapies for emotional disorders

Carmen Schaeuffele ⓘ [1,7] ✉, Laura E. Meine ⓘ [2,3,7] ✉, Ava Schulz[2,3,7],
Maxi C. Weber[1], Angela Moser[2,3], Christina Paersch[2,3,4], Dominique Recher ⓘ [2,3],
Johanna Boettcher[5], Babette Renneberg[1], Christoph Flückiger[6] & Birgit Kleim[2,3]

Transdiagnostic cognitive behavioural psychotherapy (TD-CBT) may facilitate the treatment of emotional disorders. Here we investigate short- and long-term efficacy of TD-CBT for emotional disorders in individual, group and internet-based settings in randomized controlled trials (PROSPERO CRD42019141512). Two independent reviewers screened results from PubMed, MEDLINE, PsycINFO, Google Scholar, medRxiv and OSF Preprints published between January 2000 and June 2023, selected studies for inclusion, extracted data and evaluated risk of bias (Cochrane risk-of-bias tool 2.0). Absolute efficacy from pre- to posttreatment and relative efficacy between TD-CBT and control treatments were investigated with random-effects models. Of 56 identified studies, 53 (6,705 participants) were included in the meta-analysis. TD-CBT had larger effects on depression ($g = 0.74$, 95% CI = 0.57–0.92, $P < 0.001$) and anxiety ($g = 0.77$, 95% CI = 0.56–0.97, $P < 0.001$) than did controls. Across treatment formats, TD-CBT was superior to waitlist and treatment-as-usual. TD-CBT showed comparable effects to disorder-specific CBT and was superior to other active treatments for depression but not for anxiety. Different treatment formats showed comparable effects. TD-CBT was superior to controls at 3, 6 and 12 months but not at 24 months follow-up. Studies were heterogeneous in design and methodological quality. This review and meta-analysis strengthens the evidence for TD-CBT as an efficacious treatment for emotional disorders in different settings.

Mental disorders are highly prevalent and show high comorbidity between them. Disorders across different diagnostic categories share commonalities and underlying processes on cognitive, neuropsychological and genetic levels[1–3]. Transdiagnostic cognitive behavioural therapy (TD-CBT) as an umbrella term encompasses different treatment approaches to tackle comorbidity[4]. In unified transdiagnostic treatments, patients with different disorders receive the same 'broadband' treatment that targets shared commonalities between these disorders.

[1]Department of Education and Psychology, Freie Universitaet Berlin, Berlin, Germany. [2]Experimental Psychopathology and Psychotherapy, Department of Psychology, University of Zurich, Zurich, Switzerland. [3]Department of Psychiatry, Psychotherapy and Psychosomatics, Psychiatric University Hospital Zurich, Zurich, Switzerland. [4]University Hospital of Child and Adolescent Psychiatry and Psychotherapy, University of Bern, Bern, Switzerland. [5]Clinical Psychology and Psychotherapy, Psychologische Hochschule Berlin, Berlin, Germany. [6]Department of Psychology, University of Kassel, Kassel, Germany. [7]These authors contributed equally: Carmen Schaeuffele, Laura E. Meine, Ava Schulz. ✉e-mail: carmen.schaeuffele@fu-berlin.de; laura.meine@bli.uzh.ch

**Fig. 1 | PRISMA flowchart of the literature search and screening procedure.** Three studies could not be included in the meta-analysis because either no self-report of anxiety or depression was available[29] or no data were available[27,28]. For one study[106], treatment effects at 12 months follow-up were reported in a separate publication[118] which was not included in the final number of studies as this reflects the number of RCTs identified. However, we included the follow-up values in our meta-analysis.

Examples of this approach include the unified protocol (UP) for emotional disorders[5], the anxiety treatment protocol[6] or transdiagnostic behaviour therapy[7]. Typically, unified transdiagnostic interventions apply the same selection and sequence of modules to all patients, independent of their characteristics. In tailored interventions, patients receive a treatment that is personalized to them. Different approaches to tailoring exist, from tailoring unified transdiagnostic treatments by personalizing the sequence of modules based on baseline characteristics to using idiographic case formulation to aggregate methods across different treatment packages. The key difference between unified transdiagnostic treatments and tailored interventions lies in their scope and focus. Unified transdiagnostic treatments have been specifically developed to target comorbidity by addressing shared mechanisms across different disorders in a comprehensive manner that is applicable to a range of patients. Tailored interventions, on the

other hand, focus on addressing the specific needs and characteristics of individuals. Thus, unified transdiagnostic treatments offer a broad, overarching approach that can be applied to many disorders, while tailored interventions provide a more individualized treatment approach that considers the unique aspects of each person's condition.

TD-CBT is a highly relevant approach to addressing treatment gaps and disseminating evidence-based treatments. Unified transdiagnostic approaches are especially promising in health care systems with notable treatment gaps, for example, by offering one transdiagnostic approach for emotional disorders instead of several disorder-specific approaches. In addressing a broader range of psychopathology, unified TD-CBT specifically may provide a more comprehensive treatment for patients, facilitate clinical training[8] and lower treatment costs by reducing time invested by patients and therapists[9–11]. It can also be flexibly adapted to various treatment settings, ranging from

**Table 1 | RCTs investigating TD-CBT for emotional disorders in individual, group and internet-based format**

| Authors | Country | Sample | Included diagnoses | TD-CBT protocol | Number of TD-CBT sessions | Control group(s) | Relevant measures (anxiety and depression) | Assessment times | Attrition (%)[a] |
|---|---|---|---|---|---|---|---|---|---|
| **Individual treatment format** | | | | | | | | | |
| Ref. 70 | Iran | N=40; mean age 22yr; female 85% | PD, SAD, GAD, OCD, PTSD, MDD | UP, n=13 | 13–16 | Other: CS-CBT, n=13; WLC, n=14 | BAI, BDI-II | Pre, post, 3MFU, 6MFU | UP: 7% post, 7% 3MFU, 7% 6MFU; CS-CBT: 7% post, 7% 3MFU, 7% 6MFU; WLC: 0% post, 0% 3MFU, 0% 6MFU |
| Ref. 32 | United States | N=223; mean age 31yr; female 56% | PA, GAD, OCD, SAD | UP, n=88 | 16 | DS-CBT, n=91; WLC, n=44 | OASIS, ODSIS | Pre, post, 6MFU | UP: 26% post, 31% 6MFU; DS-CBT: 31% post, 34% 6MFU; WLC: 27% post, NA 6MFU |
| Ref. 71 | Spain | N=102; mean age 38yr; female 88% | Somatoform disorder, depression disorder, PD, GAD | Brief individual psychotherapy, n=34; brief group psychotherapy, n=34 | 8 | TAU, n=34 | GAD-7, PHQ-9 | Pre, post | Brief individual: 18% post; brief group: 6% post; TAU: 12% post |
| Ref. 72 | United States | N=29; mean age 44yr; female 59% | Bipolar disorder, comorbid anxiety | UP+TAU, n=13 | 18 | TAU (pharmacotherapy), n=16 | ASI, QIDS | Pre, post | UP+TAU: 39% post; TAU: 38% post |
| Ref. 73 | United States | N=37; mean age 30yr; female 59% | GAD, SAD, OCD, PA | UP, n=26 | 18 | WLC, n=11 | BAI, BDI-II | Pre, post, 6MFU | UP: 15% post 6MFU not reported; WLC: 9% post, 6MFU not reported |
| Ref. 74 | United States | N=93; mean age 43yr; female 24% | GAD, PA, PTSD, OCD, SAD, MDD, persistent depressive disorder | TBT, n=46 | 12 | Other: BA, n=47 | DASS-A, DASS-D | Pre, post, 6MFU | TBT: 37% post, 57% 6MFU; BA: 55% post, 72% 6MFU |
| Ref. 75 | United States | N=37; mean age 47yr; female 19% | Adjustment disorder, GAD, MDD, persistent depressive disorder, PD, SAD, PTSD, SUD, other-specified trauma and stress-related disorder, other-specified depressive disorder | UP, n=13 | 12 | TAU, n=11; other: present centred therapy, n=13 | OASIS, PHQ-9 | Pre, post, 3MFU | UP: 0% post, 0% 3MFU; TAU: 0% post, 0% 3MFU; present centred therapy: 15% post, 15% 3MFU |
| Ref. 29 | Japan | N=104; mean age 37yr; female 61% | MDD, DD, PD with AG, AG without history of PD, SAD, GAD, OCD, PTSD | UP+TAU, n=52 | 12–16 | WLC+TAU, n=52 | No self-report measure of anxiety or depression | Pre, mid, post, 5.5MFU | UP+TAU: 6% post, 8% 43WFU; WLC+TAU: 10% post, 17% 43WFU |
| Ref. 76 | Iran | N=24; mean age 23yr; female 79% | OCD, GAD, SAD, PD, MDD | UP, n=12 | 20 | WLC, n=12 | BAI, BDI-II | Pre, post | UP: 0% post; WLC: 0% post |
| Ref. 77 | Iran | N=23 mean age 34yr; female 65% | GAD, SAD, PA, anxiety NOS, MDD | UP, n=11 | 8 | DS-CBT, n=12 | BAI, BDI-II | Pre, post | UP: 15% post; DS-CBT: 20% post |
| Ref. 78 | Iran | N=64; mean age 27yr; female 53% | GAD, SAD, PD, MDD | UP, n=22 | 12 | WLC, n=19; other: CBT-P, n=23 | BAI, BDI-II | Pre, post, 6MFU | UP: 0% post, 0% 6MFU; CBT-P: 0% post, 0% 6MFU; WLC: 0% post, 0% 6MFU |
| Ref. 28 | Canada | N=59; mean age 30yr; female 79% | PD, SAD, GAD, SP, PTSD, anxiety NOS | RT+CBT, n=21 | 4 | WLC, n=19; RT, n=19 | GAD-7, DASS-D | Pre, 1WFU, 1MFU, 3MFU | RT+CBT: 10% 1WFU, 19% 1MFU, 43% 3MFU; RT: 0% 1WFU, 0% 1MFU, 5% 3MFU; WLC: 0% 1WFU, 16% 1MFU, 16% 3MFU |
| Ref. 79 | United States | N=53; mean age 39yr; female 75% | GAD, MDD | ERT, n=28 | 20 | MAC, n=25 | STAI-7, BDI-II | Pre, post, 3MFU, 9MFU | ERT: 11% post, 25% 3MFU, 29% 9MFU; MAC: 20% post |
| Ref. 80 | Iran | N=43; mean age 21yr; female 74% | GAD, MDD | UP, n=15; UP-tDCS, n=13 | 12 | WLC, n=15 | BAI, BDI-II | Pre, post, 3MFU | UP: 0% post, 0% 3MFU; UP-tDCS: 0% post, 0% 3MFU; WLC: 0% post, 0% 3MFU |
| Ref. 81 | United States | N=254; mean age 27yr; female 0% | MDD, DD, PD, AG, SAD, OCD, PTSD, GAD, AUD, SUD | ESTEEM, n=100 | 10 | Other: LGBQ-affirmative counselling, n=102; TAU (HIV testing and counselling), n=52 | OASIS, ODSIS | Pre, 4MFU, 8MFU, 12MFU | ESTEEM: 17% 4MFU, 18% 8MFU, 19% 12MFU; LGBQ-affirmative counselling: 25% 4MFU, 25% 8MFU, 21% 12MFU; TAU: 17% 4MFU, 19% 8MFU, 31% 12MFU |

**Table 1 (continued) | RCTs investigating TD-CBT for emotional disorders in individual, group and internet-based format**

| Authors | Country | Included diagnoses | Sample | TD-CBT protocol | Number of TD-CBT sessions | Control group(s) | Relevant measures (anxiety and depression) | Assessment times | Attrition (%)[a] |
|---|---|---|---|---|---|---|---|---|---|
| Ref. 31 | United States | PD, GAD, SAD, PTSD | N=1,004; mean age 44 yr; female 71% | CALM, n=503 | 6–8 | TAU (pharmacotherapy and/or counselling by physician), n=501 | BSI-12 subscale for anxiety, PHQ-9 | Pre, 6MFU, 12MFU, 18MFU | CALM: 11% 6MFU, 19% 12MFU, 19% 18MFU; TAU: 14% 6MFU, 20% 12MFU, 21% 18MFU |
| Ref. 27 | Iran | OCD, GAD, SAD, anxiety NOS, MDD | N=24; mean age not reported; female 83% | UP, n=12 | 20 | WLC, n=12 | BAI, BDI-II | Pre, post, 1MFU | UP: 0% post, 0% 1MFU; WLC: 0% post, 0% 1MFU |
| Ref. 82 | Australia | MDD (depressive disorder with melancholic features), PD, AG, SAD, GAD, OCD, PTSD | N=19; mean age 62 yr; female 53% | UP, n=9 | 12 | other: EUC, n=10 | GAD-7, PHQ-9 | Pre, post, 6MFU | UP: 0% post, 0% 6MFU; EUC: 0% post, 0% 6MFU |
| **Group treatment format** | | | | | | | | | |
| Ref. 83 | Germany | MDD | N=218; mean age 39 yr; female 64% | ART, n=76 | 7 | WLC, n=72; CFC, n=70 | BDI-II | Pre, post, 4WFU | ART: 32% post, 29% 4WFU; WLC: 29% post, 29% 4WFU; CFC: 40% post, 29% 4WFU |
| Ref. 84 | Spain | GAD, MDD, PD, somatization disorder | N=105; mean age 40 yr; female 69% | Brief group TD therapy, n=53 | 8 | TAU, n=52 | GAD-7, PHQ-9 | Pre, post | Brief group TD: 11% post; TAU: 19% post |
| Ref. 85 | Brazil | PA, GAD, SAD, PTSD, SP, MDD | N=67; mean age 34 yr; female 81% | UP, n=33 | 14 | TAU (pharmacotherapy), n=34 | BAI, BDI | Pre, post | UP: 27% post; TAU: 29% post |
| Ref. 86 | United Kingdom | Mood disorders, anxiety disorders, stress and somatoform disorders, eating disorders, alcohol dependence | N=235; mean age 46 yr; female 84.26% | MMI, n=80 | 12 | CBT and TAU (pharmacotherapy), n=84; TAU only, n=81 | CPRS-S-A, anxiety; CPRS-S-A, depression | Pre, post, 12MFU | MMI: 18% post, 5% 12MFU; CBT and TAU: 18% post, 4% 12MFU; TAU: 20% post, 4% 12MFU |
| Ref. 87 | Canada | SAD, PA, GAD, PTSD[69], OCD, SP | N=60; mean age 41yr; female 64% | TD group CBT for patients with various anxiety disorders, n=33 | 11 | WLC, n=27 | BAI (anxiety only) | Pre, post, 6MFU | TD group CBT: 55% post, 55% 6MFU; WLC: 66% post, 66% 6MFU |
| Ref. 88 | Spain | Depressive disorder, anxiety disorder | N=128; mean age 41yr; female 77% | TD-CBT, n=33 | 8 | WLC, n=34; other: BA, n=34; other: ACT, n=27 | GAD-7, BDI-IA | Pre, post, 3MFU, 6MFU | TD-CBT: 15% post, 18% 3MFU, 21% 6MFU; WLC: 24% post, 24% 3MFU, 24% 6MFU; BA: 29% post, 38% 3MFU, 38% 6MFU; ACT: 22% post, 26% 3MFU, 30% 6MFU |
| Ref. 89 | Germany | PTSD, MDD | N=24; mean age 22yr; female 0% | CA-CBT+, n=12 | 12 | WLC, n=12 | PHQ-9 (depression only) | Pre, post, 12MFU | CA-CBT+: 0% post, 0% 12MFU; WLC: 0% post, 0% 12MFU |
| Ref. 90 | the Netherlands | SAD, GAD, PD, AG, SP, anxiety NOS | N=129; mean age 34yr; female 71% | CBT, n=67 | 15 | Other: AET, n=62 | SCL-90 subscale for anxiety, BDI-II | Pre, post, 3MFU, 6MFU, 12MFU | CBT: 25% post, 39% 3MFU, 53% 6MFU, 63% 12MFU; AET: 18% post, 27% 3MFU, 50% 6MFU, 71% 12MFU |
| Ref. 91 | Iran | MDD, DD, GAD, SAD | N=70; mean age 35yr; female 63% | UP, n=35 | 12 | TAU, n=35 | HADS-A, HADS-D | Pre, post | UP: 20% post; TAU: 34% post |
| Ref. 92 | Iran | Depressive disorder, anxiety disorder | N=64; mean age 35yr; female 100% | UP, n=32 | 14 | TAU, n=32 | HADS-A, HADS-D | Pre, post, 3MFU | UP: 6% post, 13% 3MFU; TAU: 13% post, 28% 3MFU |
| Ref. 93 | United States | PA, GAD, SAD, SP, OCD, PTSD, anxiety NOS, MDD, DD, depression NOS, SUD | N=44; mean age 36yr; female 66% | DBT-ST, n=22 | 16 | Other: client-centred ST, n=22 | OASIS, PHQ-9 | Pre, post, 2MFU | DBT-ST: 32% post, 32% 2MFU; ST: 59% post, 59% 2MFU |
| Ref. 94 | United States | SAD, PA, GAD, anxiety NOS, OCD, SP | N=85; mean age 33yr; female 62% | TD group CBT, n=64 | 12 | Other: comprehensive relaxation training programme, n=21 | BAI (anxiety only) | Pre, post | TD group CBT: 36% post relaxation, 62% post |
| Ref. 95 | United States | SAD, GAD, PA | N=46; mean age 31yr; female 50% | TD group CBT, n=23 | 12 | DS-CBT, n=23 | STAI-S, BDI-II | Pre, post | TD group CBT: 30% post; DS-CBT: 48% post |

**Table 1 (continued) | RCTs investigating TD-CBT for emotional disorders in individual, group and internet-based format**

| Authors | Country | Included diagnoses | Sample | TD-CBT protocol | Number of TD-CBT sessions | Control group(s) | Relevant measures (anxiety and depression) | Assessment times | Attrition (%)[a] |
|---|---|---|---|---|---|---|---|---|---|
| Ref. 6 | United States | SAD, OCD, GAD, PA, PTSD | N=19; mean age 40yr; female 61% | TD group CBT, n=9 | 12 | WLC, n=10 | DASS-A DASS-D[b] | Pre, post | TD group CBT: 30% post; WLC:10% post |
| Ref. 96 | Spain | GAD, PA, AG, OCD, PTSD, SAD, hypochondria, anxiety NOS, MDD, DD, unspecified mood disorder | N=243; mean age 43yr; female 79% | UP, n=131 | 12 | TAU, n=112 | PDSS, BDI-II | Pre, post, 3MFU, 6MFU | UP: 34% post, 40% 3MFU, 51% 6MFU; TAU: 28% post, 39% 3MFU, 59% 6MFU |
| Ref. 97 | Spain | GAD, PA, AG, OCD, PTSD, SAD, hypochondria, anxiety NOS, MDD, DD, unspecified mood disorder | N=488; mean age 43yr; female 79% | UP, n=279 | 12 | TAU, n=209 | BAI, BDI-II | Pre, post, 3MFU, 6MFU | UP: 35% post, 43% 3MFU, 54% 6MFU; TAU: 28% post, 40% 3MFU, 52% 6MFU |
| Ref. 33 | Denmark | MDD, SAD, PD, AG | N=191; mean age 32yr; female 65% | UP, n=98 | 14 | DS-CBT, n=93 | HAM-A, HAM-D | Pre, post, 6MFU | UP: 0% post, 0% 6MFU; DS-CBT: 0% post, 0% 6MFU |
| Ref. 98 | Canada | GAD, SAD, PD, AG | N=231; mean age 37yr; female 86% | TD-CBT + TAU, n=117 | 12 | TAU, n=114 | BAI, PHQ-9 | Pre, post, 4MFU, 8MFU | TD-CBT: 19% post; TAU: 7% post |
| Ref. 99 | Belgium | GAD, MDD | N=80; mean age 43yr; female 66% | RNT-G, n=45 | 8 | WLC, n=35 | STAI-S, BDI-II | Pre, post, 3MFU, 9MFU | RNT-G: 7% post, 7% 3MFU, 7% 9MFU; WLC: 14% post |
| Ref. 46 | United States | PA, GAD, SAD | N=96; mean age 36yr; female 72% | F-SET, n=57 | 10 | WLC, n=39 | SPRAS, BDI-II | Pre, post, 6MFU | F-SET: 7% post, 26% 6MFU; WLC: 0% post, NA 6MFU |
| Ref. 100 | Iran | GAD, SAD, PD | N=43; mean age 23yr; female 67% | UP, n=20 | 14 | WLC, n=23 | BAI, BDI-II | Pre, post, 3MFU | UP: not reported; WLC: not reported |
| **Internet-based treatment format** | | | | | | | | | |
| Ref. 101 | Switzerland, Germany and Austria | SAD, PA, GAD | N=139; mean age 42yr; female 71% | Velibra, unguided (TD-iCBT+TAU), n=70 | 6 | TAU (everything; no restrictions), n=69 | BAI, BDI-II | Pre, post, 6MFU | Velibra: 19% post, 37% 6MFU; TAU: 9% post, NA 6MFU |
| Ref. 102 | Australia | GAD | N=338; mean age 44yr; female 76% | Well-being course, guided and unguided (TD-iCBT), n=170 | 5 | DS-iCBT for GAD, guided and unguided, n=168 | GAD-7, PHQ-9 | Pre, post, 3MFU, 12MFU, 24MFU | Well-being course: 16% post, 21% 3MFU, 25% 12MFU, 26% 24MFU; DS-iCBT: 17% post, 15% 3MFU, 21% 12MFU, 20% 24MFU |
| Ref. 103 | Australia | SAD | N=220; mean age 42yr; female 58% | Well-being course, guided and unguided (TD-iCBT), n=105 | 5 | DS-iCBT for SAD, guided and unguided, n=115 | MINI-SPIN, PHQ-9 | Pre, post, 3MFU, 12MFU, 24MFU | Well-being course: 21% post, 22% 3MFU, 27% 12MFU, 22% 24MFU; DS-iCBT: 23% post, 28% 3MFU, 23% 12MFU, 23% 24MFU |
| Ref. 104 | Spain | MDD, DD, GAD, PD/AG, PD, AG, SAD, OCD, anxiety NOS, depression NOS | N=216; mean age 34yr; female 72% | TIBP, unguided, n=7; TIBP+positive affect, unguided, n=73 | 18 | WLC, n=72 | BAI, BDI-II | Pre, post | TIBP: 37% post; TIBP+positive affect: 37% post; WLC: 24% post |
| Ref. 105 | Australia | PA | N=145; mean age 41yr; female 79% | Well-being course, guided and unguided (TD-iCBT), n=72 | 5 | DS-iCBT for PA, guided and unguided, n=73 | PDSS, PHQ-9 | Pre, post, 3MFU, 12MFU, 24MFU | Well-being course: 11% post, 12% 3MFU, 18% 12MFU, 24% 24MFU; DS-iCBT: 20% post, 20% 3MFU, 26% 12MFU, 23% 24MFU |
| Ref. 106 | Spain | GAD, AG, PD, SAD, OCD, MDD, DD, anxiety NOS, depression NOS | N=200; mean age 38yr; female 69% | Emotion regulation, guided (TIBP), n=99 | 18 | TAU, n=101 | BAI, BDI-II | Pre, post, 3MFU, 12MFU[c] | Emotion regulation: 36% post, 48% 3MFU, 57% 12MFU; TAU: 34% post, 45% 3MFU, 56% 12MFU |
| Ref. 107 | Australia | GAD, SAD, PA | N=131; mean age 42yr; female 59% | Anxiety programme, guided (TD-iCBT), n=89 | 8 | WLC, n=42 | GAD-7, PHQ-9 | Pre, post, 3MFU | Anxiety programme: 9% post, 17% 3MFU; WLC: 2% post, 19% 3MFU |

**Table 1 (continued) | RCTs investigating TD-CBT for emotional disorders in individual, group and internet-based format**

| Authors | Country | Included diagnoses | Sample | TD-CBT protocol | Number of TD-CBT sessions | Control group(s) | Relevant measures (anxiety and depression) | Assessment times | Attrition (%)[a] |
|---|---|---|---|---|---|---|---|---|---|
| Ref. 108 | Australia | MDD, GAD, SAD, PD, AG, OCD | N=158; mean age 39 yr; female 86% | iCBT, guided, n=39; ME-iCBT, guided, n=40 | 6 | TAU, n=39; iMT, guided, n=40 | GAD-7, PHQ-9 | Pre, post, 3MFU | iCBT: 23% post, 33% 3MFU; ME-iCBT: 33% post, 28% 3MFU; TAU: 15% post, NA 3MFU; iMT: 38% post, 48% 3MFU |
| Ref. 109 | China | Anxiety disorders, depressive disorders, other emotion-related disorders | N=75; mean age 32 yr; female 71% | iMIED+TAU, n=37 | 8 | TAU, n=38 | BAI, BDI-II | Pre, post | iMIED+TAU: 22% post; TAU: 5% post |
| Ref. 110 | Australia | GAD, PA, SAD, MDD | N=53; mean age 28 yr; female 64% | UniWellbeing course, guided (TD-iCBT for students), n=30 | 5 | WLC, n=23 | GAD-7, PHQ-9 | Pre, post, 3MFU | UniWellbeing: 30% post, 40% 3MFU; WLC: 9% post, NA 3MFU |
| Ref. 111 | Australia | GAD, MDD | N=100; mean age 44 yr; female 78% | Worry and sadness programme, guided (TD-iCBT), n=46 | 6 | WLC, n=54 | GAD-7, PHQ-9 | Pre, post, 3MFU | Worry and sadness: 7% post, 13% 3MFU; WLC: 2% post, NA 3MFU |
| Ref. 112 | Germany | AG, GAD, PD, SAD, MDD, persistent depressive disorder, somatic symptom disorder, illness anxiety disorder | N=129; mean age 37 yr; female 68% | Internet-based UP, guided, n=65 | 10 | WLC, n=64 | GAD-7, PHQ-9 | Pre, post | UP: 35% post; WLC: 11% post |
| Ref. 113 | Australia | GAD, SAD, PA | N=78; mean age 40 yr; female 68% | Anxiety programme, guided (TD-iCBT), n=40 | 6 | WLC, n=38 | GAD-7, PHQ-9 | Pre, post, 3MFU | TD: 10% post, 20% 3MFU; WLC: 5% post, NA 3MFU |
| Ref. 114 | Australia | GAD, PA, SAD, MDD | N=74; mean age 44 yr; female 73% | Well-being programme, guided (TD-iCBT), n=37 | 8 | WLC, n=37 | GAD-7, PHQ-9 | Pre, post, 3MFU | TD: 8% post, 14% 3MFU; WLC: 5% post, NA 3MFU |
| Ref. 115 | Australia | MDD | N=290; mean age 44 yr; female 72% | Well-being course, guided (TD-iCBT), n=149 | 5 | DS-iCBT for MDD, n=142 | GAD-7, PHQ-9 | Pre, post, 3MFU, 12MFU, 24MFU | TD: 5% post, 20% 3MFU, 30% 12MFU, 24% 24MFU; DS-iCBT: 16% post, 16% 3MFU, 20% 12MFU, 18% 24MFU |
| Ref. 116 | Romania | GAD, SAD, PA, PTSD, SP, OCD, NOS, MDD | N=97; mean age 34 yr; female 81% | Internet-based UP, guided, n=64 | 9 | WLC, n=33 | OASIS, BDI-II | Pre, post, 6MFU | UP: 22% post, 45% 3MFU; WLC: 6% post, NA 6MFU |
| Ref. 117 | Afghanistan | Depressive disorder, anxiety disorder | N=102; mean age 28 yr; female 47% | Internet-based UP, unclear whether guided or unguided, n=51 | 12–14 | TAU, n=51 | OASIS, ODSIS | Pre, post | UP: 22% post; TAU: 39% post |

ACT, acceptance and commitment therapy; AET, autonomy enhancing therapy; AG, agoraphobia; ART, affect regulation training; ASI, anxiety sensitivity index; AUD, alcohol use disorder; BA, behavioural activation; BDI, Beck depression inventory; BSI, brief symptom inventory; CALM, coordinated anxiety learning and management; CA-CBT+, culturally adapted cognitive behaviour therapy plus problem management; CBT-P, CBT for perfectionism; CFC, common factor control; CPRS-S-A, self-rating-scale for affective syndromes; CS, construct-specific; DASS, short form of depression, anxiety and stress scale; DBT-ST, dialectical behaviour therapy skills training; DD, dysthymic disorder; DS, disorder-specific; ERT, emotion regulation therapy; ESTEEM, effective skills to empower effective men; EUC, enhanced usual care; F-SET, false safety behaviour elimination therapy; HADS, hospital anxiety and depression scale; HAM-A, Hamilton anxiety rating scale; HAM-D, Hamilton depression rating scale; iCBT, internet-based CBT; iMIED, internet-based self-help mindfulness intervention for emotional distress; iMT, mindfulness training; MAC, modified attention control; ME-iCBT, mindfulness-enhanced iCBT; MFU, month follow-up; MINI-SPIN, mini-social phobia inventory; MMI, multimodal intervention; NOS, not otherwise specified; OASIS, overall anxiety severity and impairment scale; OCD, obsessive compulsive disorder; ODSIS, overall depression severity and impairment scale; PA, panic disorder with/without agoraphobia; PD, panic disorder; PDSS, panic disorder severity scale; PHQ, patient health questionnaire; PTSD, post-traumatic stress disorder; QIDS, quick inventory of depressive symptomatology; RNT-G, group treatment for repetitive negative thinking; RT, resistance training; SCL, symptom checklist; SP, specific phobias; SPRAS, Sheehan patient-rated anxiety inventory—state; SUD, substance use disorder; TBT, transdiagnostic behaviour therapy; tDCS, transcranial direct current stimulation; TIBP, transdiagnostic internet-based protocol; UP, unified treatment protocol for emotional disorders; WFU, week follow-up; WLC, waitlist control. [a]Missing data as a percentage of randomized individuals who did not provide further assessments. [b]Data on depression-related outcomes (DASS-D) were provided by the authors upon request. [c]Data for 12MFU were provided in ref. 118.

individual and group face-to-face formats, to scalable internet-based self-help formats[12,13].

Several reviews and meta-analyses investigated TD treatments. However, these previous meta-analyses differed in the settings they investigated (group, individual or internet-based), the target population (anxiety, anxiety and depression or emotional disorders) and the breadth of the transdiagnostic definition they applied (unified TD approaches, tailored interventions or specific treatment protocols). Others[14] focused on face-to-face treatments in anxiety disorders and another meta-analysis[15] compared TD treatments to disorder-specific treatments. Two meta-analyses[16,17] investigated unified TD and tailored interventions in the internet-based setting and another[18] focused on TD interventions in group format. Others, such as ref. 19, focused on a specific TD treatment, the UP[5]. The most recent meta-analysis[20] aggregated findings from their large database on treatments for depression (https://www.metapsy.org/) that had a transdiagnostic stance. However, the authors did not focus exclusively on CBT and did not include transdiagnostic treatments targeting anxiety or other emotional disorders, although most transdiagnostic treatments are aimed at anxiety disorders. Two studies[21,22] are the most comprehensive meta-analyses on TD-CBT for emotional disorders to date, including all settings as well as a focus on anxiety and depression. However, the search conducted by ref. 21 ended in 2013 and—given the novelty of TD-CBT then—they could only include four randomized controlled trials (RCTs) in their meta-analysis. On the other hand, ref. 22 only included clinician-guided internet-based interventions and did not restrict their study selection to RCTs. Thus, self-guided internet-based treatments without clinician support have not been included in their review, although it is a scalable format which has a particularly strong potential to reach larger populations.

Overall, unified TD treatments seem to produce large pre- to posttreatment effects in different settings. However, several questions need to be addressed: the comparability of unified TD-CBT to disorder-specific treatments, as there is conflicting meta-analytic evidence[15,22], the comparability across different settings and the long-term effects. The surge in research activity on TD protocols in recent years warrants an updated comprehensive review and meta-analysis that aggregates findings across different unified TD protocols and settings. The current review and meta-analysis expands previous reviews and meta-analyses by investigating unified TD-CBT for emotional disorders in individual and group face-to-face settings, including internet-based interventions with and without clinician guidance.

We focused on treatments based on CBT principles and unified approaches (excluding tailored treatments), to (1) update results on short- and long-term efficacy of TD-CBT and (2) compare effects of transdiagnostic protocols to different types of control conditions, including waitlist control, treatment-as-usual (TAU), disorder-specific CBT (DS-CBT) and other active interventions. To summarize: in adult patients with emotional disorders (population), what is the effect of TD-CBT (intervention) on anxiety and depression (outcome) compared with waitlist, TAU, DS-CBT and other active interventions (comparison) at posttreatment and follow-ups?

## Results

### Included studies

Figure 1 shows the preferred reporting items for systematic reviews and meta-analyses (PRISMA) flowchart of the literature search and screening procedure. By systematic search and screening, we identified 56 eligible RCTs, including 6,916 individuals, published between 2005 and 2023. No preprints could be included in the final selection. Half the included studies were published after 2019 and RCTs with internet-based treatment format date from 2010 and later. Table 1 summarizes characteristics of the individual studies.

Most of the studies were conducted in Europe (n = 16), the United States (n = 13) and Australia (n = 11). We could also include several RCTs

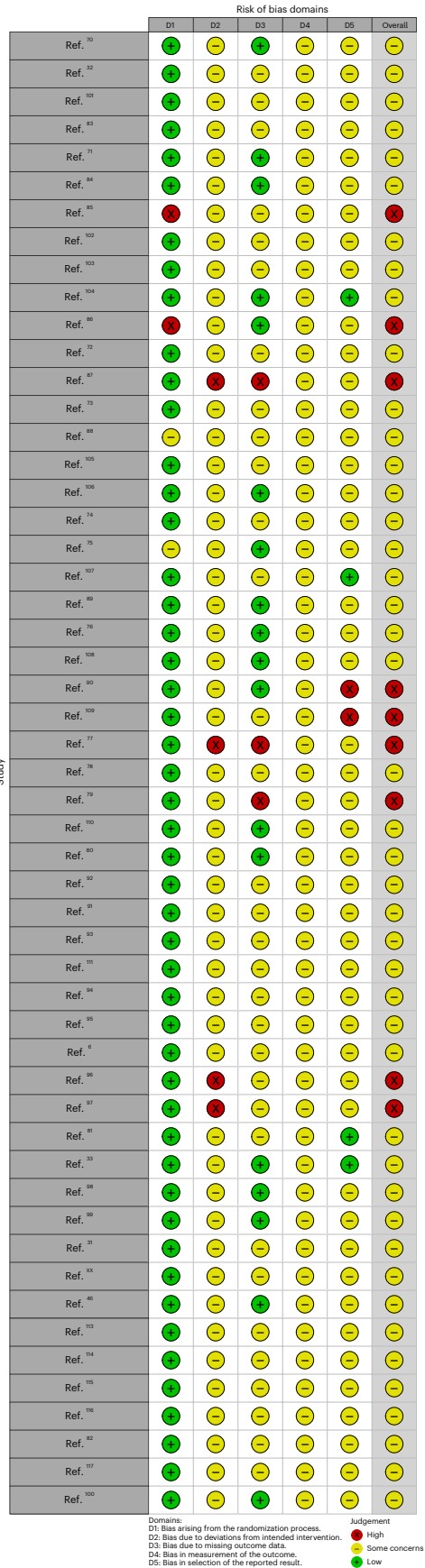

**Fig. 2 | Risk of bias assessment.** Traffic-light plot of the domain-level judgements. Risk of bias was assessed across five domains for each study included in the meta-analysis using the revised Cochrane risk-of-bias tool (RoB 2.0). The combination of assessments in the five domains results in an overall risk of bias rating.

**Table 2 | Between-group effect sizes of depressive and anxiety symptoms for transdiagnostic treatments compared to control groups at posttreatment**

| | | Depression | | | | | | | | Anxiety | | | | | | | |
|---|---|---|---|---|---|---|---|---|---|---|---|---|---|---|---|---|---|
| | | *k* | *g* | *P* | LL | UL | *I²* | *Q* | *P* | *k* | *g* | *P* | LL | UL | *I²* | *Q* | *P* |
| **TD-CBT (all treatment formats) versus** | Control | 63 | 0.74 | <0.001 | 0.57 | 0.92 | 88.29 | 344.96 | <0.001 | 61 | 0.77 | <0.001 | 0.56 | 0.97 | 91.72 | 398.55 | <0.001 |
| | DS-CBT | 9 | 0.09 | 0.269 | -0.07 | 0.25 | 53.96 | 17.82 | 0.023 | 9 | 0.09 | 0.091 | −0.01 | 0.20 | 5.79 | 12.56 | 0.128 |
| | TAU | 18 | 0.90 | <0.001 | 0.66 | 1.14 | 77.28 | 61.55 | <0.001 | 17 | 0.98 | <0.001 | 0.63 | 1.33 | 89.75 | 101.68 | <0.001 |
| | Other | 13 | 0.27 | <0.001 | 0.13 | 0.42 | 0.010 | 13.76 | 0.316 | 13 | 0.14 | 0.128 | −0.04 | 0.31 | 17.67 | 15.33 | 0.224 |
| | WL | 23 | 1.23 | <0.001 | 0.80 | 1.66 | 92.85 | 124.8 | <0.001 | 22 | 1.24 | <0.001 | 0.82 | 1.67 | 92.26 | 122.48 | <0.001 |
| **Individual TD-CBT versus** | Control | 18 | 0.90 | <0.001 | 0.57 | 1.23 | 75.88 | 68.17 | <0.001 | 17 | 1.09 | <0.001 | 0.62 | 1.56 | 87.16 | 98.86 | <0.001 |
| | DS-CBT | – | – | – | – | – | – | – | – | – | – | – | – | – | – | – | – |
| | TAU | 4 | 1.08 | <0.001 | 0.73 | 1.43 | 0 | 3.11 | 0.374 | 3 | 1.33 | 0.002 | 0.49 | 2.16 | 74.24 | 7.23 | 0.027 |
| | Other | 5 | 0.49 | 0.011 | 0.11 | 0.86 | 25.73 | 5.99 | 0.200 | 5 | 0.25 | 0.187 | −0.12 | 0.62 | 26.40 | 5.28 | 0.259 |
| | WL | 7 | 1.40 | <0.001 | 0.87 | 1.93 | 71.24 | 23.89 | <0.001 | 7 | 1.71 | <0.001 | 0.87 | 2.55 | 87.80 | 34.67 | <0.001 |
| **Group-based TD-CBT versus** | Control | 24 | 0.87 | <0.001 | 0.41 | 1.32 | 96.09 | 167.38 | <0.001 | 23 | 0.76 | <0.001 | 0.32 | 1.20 | 95.56 | 188.36 | <0.001 |
| | DS-CBT | 3 | 0.16 | 0.307 | −0.15 | 0.48 | 59.79 | 4.98 | 0.083 | 3 | 0.14 | 0.331 | −0.14 | 0.41 | 49.59 | 3.69 | 0.158 |
| | TAU | 8 | 0.94 | <0.001 | 0.49 | 1.40 | 88.94 | 39.16 | <0.001 | 8 | 1.04 | 0.003 | 0.36 | 1.73 | 95.08 | 68.66 | <0.001 |
| | Other | 6 | 0.23 | 0.017 | 0.04 | 0.41 | 0.010 | 5.50 | 0.358 | 6 | 0.07 | 0.669 | −0.24 | 0.37 | 49.27 | 9.49 | 0.091 |
| | WL | 7 | 1.88 | 0.027 | 0.21 | 3.55 | 98.14 | 84.02 | <0.001 | 6 | 1.52 | 0.033 | 0.12 | 2.92 | 96.85 | 66.10 | <0.001 |
| **Internet-based TD-CBT versus** | Control | 21 | 0.61 | <0.001 | 0.42 | 0.80 | 78.45 | 101.63 | <0.001 | 21 | 0.58 | <0.001 | 0.39 | 0.78 | 79.16 | 97.35 | <0.001 |
| | DS-CBT | 4 | 0.08 | 0.563 | −0.18 | 0.33 | 70.37 | 11.37 | 0.010 | 4 | 0.03 | 0.662 | −0.11 | 0.17 | 0 | 0.47 | 0.925 |
| | TAU | 6 | 0.79 | <0.001 | 0.46 | 1.12 | 69.03 | 15.73 | 0.008 | 6 | 0.76 | <0.001 | 0.43 | 1.09 | 69.15 | 16.09 | 0.007 |
| | Other | – | – | – | – | – | – | – | – | – | – | – | – | – | – | – | – |
| | WL | 9 | 0.86 | <0.001 | 0.72 | 1.01 | 0 | 7.89 | 0.444 | 9 | 0.83 | <0.001 | 0.66 | 1.00 | 20.87 | 11.07 | 0.198 |

*k*=number of comparisons; LL=lower limit of 95% CI; UL=upper limit of 95% CI. *I²* values are reported as percentage (%). Comparisons with *k*<3 studies are not reported.

from Iran (*n* = 9) and other countries. Samples investigated ranged in size from 19 to 1,004 participants (median = 94.5) and were mainly comprised of females (*M* = 66%, s.d. = 19%), with a median age of 37 years. Most frequent among included diagnoses were generalized anxiety disorder (GAD; 79%), social anxiety disorder (SAD; 70%) and major depressive disorder (MDD; 55%). Most studies investigated the UP (*n* = 23) or similar treatments. Supplementary Table 1 gives an overview of the TD-CBT protocols we included in our review. The mean number of sessions was 11.19 (s.d. = 4.32), with a range of 4–20 sessions. TD-CBT was most frequently compared to a waitlist condition (*n* = 25), followed by TAU (*n* = 18), other active treatments (*n* = 15) and DS-CBT (*n* = 8). Some RCTs included many comparison groups. Treatments were mainly carried out in a group setting (*n* = 21), followed by individual formats (*n* = 18) and internet-based approaches (*n* = 17). Self-report questionnaires on symptoms of anxiety and depression were included in nearly all studies, except for five (one included no self-reports, two assessed only anxiety and another two only depression). Most RCTs used many questionnaires. The Beck anxiety inventory (BAI) (*k* = 17) (ref. 23) and generalized anxiety disorder screener (GAD-7) (*k* = 14) (ref. 24) were most commonly used to measure anxiety and the Beck depression inventory (BDI-II) (*k* = 21) (ref. 25) and patient health questionnaire (PHQ-9) (*k* = 19) (ref. 26) to assess depression. Around 75% of RCTs included at least one follow-up assessment (typically at 3 or 6 months), allowing for the investigation of longer-term effectiveness. Sixteen studies included a second follow-up (mostly at 6 or 12 months) and eight reassessed participants for a third time (mostly at 24 months). Attrition rates at posttreatment in TD-CBT samples were similar compared to control condition samples but differed by treatment format (individual setting: *M* = 12%, s.d. = 13%; group setting: *M* = 21%, s.d. = 14%; internet-based setting: *M* = 20%, s.d. = 10%).

For the meta-analytic calculations, we excluded refs. 27,28 because the data were not available and ref. 29 because no self-report of anxiety

or depression was available, which resulted in a total *N* = 6,705 individuals for the meta-analytic calculations.

**Risk of bias assessment**

Agreement between the two independent raters in coding the risk of bias criteria was strong (*M* = 90.31%, s.d. = 7.82%, range 73.58–100%). Instances of disagreement mainly reflect differing levels of a rating; for example, 'yes' versus 'probably yes' but not the general direction. All ratings and the code for analysis of percentage agreement can be found on the Open Science Framework repository (see Data availability and Code availability). Figure 2 provides an overview of the risk of bias assessment for the five domains rated (see the section on 'Study quality assessment') for the individual studies. In Supplementary Fig. 1, we also provide a summary plot, depicting the percentage of studies showing low/high risk of bias or some concerns in each domain. We found that, overall, the risk of bias assessment of most of the included studies showed some concerns and no study was free from any risk of bias. Although there were hardly any concerns about bias in the randomization process, all studies showed some concerns for blinding of therapists, as they needed to be aware of the protocol they were providing, and assessors because we only included self-report outcomes. While intention-to-treat analyses were conducted in most studies, few reported comprehensive tests of potential bias in results due to missing outcome data, raising some concerns. Finally, although trial registrations were available for almost all included RCTs, hardly any studies provided an a priori specified analysis plan.

**Meta-analysis**

**Controlled effect sizes.** Tables 2 and 3 show controlled effect sizes as well as measures of heterogeneity (*Q* statistic and *I²*) for depression and anxiety outcomes for individual, group or internet-based settings, comparing TD-CBT to DS-CBT, TAU, waitlist and other treatments

**Table 3 | Between-group effect sizes of depressive and anxiety symptoms for transdiagnostic treatments compared to control groups at follow-up**

| | | | Depression | | | | | | | Anxiety | | | | | | |
|---|---|---|---|---|---|---|---|---|---|---|---|---|---|---|---|---|
| | | | k | g | P | LL | UL | I² | Q | P | k | g | P | LL | UL | I² | Q | P |
| **3 month FU** | **TD-CBT (all treatment formats) versus** | Control | 29 | 0.55 | <0.001 | 0.30 | 0.80 | 89.98 | 143.19 | <0.001 | 24 | 0.48 | 0.002 | 0.18 | 0.79 | 92.31 | 131.28 | <0.001 |
| | | DS-CBT | 5 | 0.11 | 0.376 | −0.14 | 0.37 | 77.81 | 15.73 | 0.003 | 5 | −0.01 | 0.912 | −0.18 | 0.16 | 52.62 | 8.48 | 0.075 |
| | | TAU | 6 | 0.42 | <0.001 | 0.26 | 0.59 | 6.45 | 6.77 | 0.238 | 4 | 0.49 | <0.001 | 0.22 | 0.76 | 51.43 | 6.24 | 0.101 |
| | | Other | 10 | 0.33 | 0.014 | 0.07 | 0.59 | 57.15 | 20.99 | 0.013 | 8 | 0.24 | 0.009 | 0.06 | 0.42 | 0 | 9.69 | 0.207 |
| | | WL | 8 | 1.29 | 0.002 | 0.47 | 2.10 | 93.43 | 70.96 | <0.001 | 7 | 1.31 | 0.016 | 0.24 | 2.37 | 94.84 | 77.31 | <0.001 |
| | **Individual TD-CBT versus** | Control | 8 | 1.38 | <0.001 | 0.64 | 2.13 | 90.69 | 51.37 | <0.001 | 5 | 1.88 | 0.002 | 0.72 | 3.03 | 91.25 | 50.83 | <0.001 |
| | **Group-based TD-CBT versus** | Control | 12 | 0.39 | 0.002 | 0.14 | 0.63 | 74.49 | 31.78 | <0.001 | 10 | 0.42 | <0.001 | 0.20 | 0.65 | 61.80 | 22.45 | 0.008 |
| | **Internet-based TD-CBT versus** | Control | 9 | 0.18 | 0.127 | −0.05 | 0.42 | 73.17 | 27.64 | <0.001 | 9 | −0.05 | 0.409 | −0.17 | 0.07 | 2.55 | 7.89 | 0.444 |
| **6 month FU** | **TD-CBT (all treatment formats) versus** | Control | 17 | 0.20 | <0.001 | 0.10 | 0.30 | 17.75 | 24.09 | 0.088 | 19 | 0.23 | <0.001 | 0.11 | 0.36 | 42.99 | 34.74 | 0.010 |
| | | DS-CBT | 3 | −0.01 | 0.937 | −0.23 | 0.21 | 39.89 | 3.28 | 0.194 | 3 | 0.04 | 0.650 | −0.15 | 0.24 | 22.99 | 2.33 | 0.312 |
| | | TAU | 4 | 0.26 | <0.001 | 0.15 | 0.38 | 0 | 0.74 | 0.864 | 4 | 0.32 | 0.001 | 0.13 | 0.52 | 47.22 | 5.67 | 0.129 |
| | | Other | 7 | 0.15 | 0.163 | −0.06 | 0.37 | 10.92 | 6.13 | 0.409 | 8 | 0.19 | 0.050 | 0 | 0.38 | 0 | 6.19 | 0.518 |
| | | WL | 3 | 0.63 | 0.019 | 0.10 | 1.15 | 47.71 | 3.82 | 0.148 | 4 | 0.62 | 0.065 | −0.04 | 1.27 | 77.44 | 11.90 | 0.008 |
| | **Individual TD-CBT versus** | Control | 9 | 0.24 | 0.012 | 0.05 | 0.44 | 44.45 | 14.07 | 0.080 | 10 | 0.28 | <0.001 | 0.18 | 0.38 | 0 | 18.92 | 0.026 |
| | **Group-based TD-CBT versus** | Control | 8 | 0.16 | 0.033 | 0.01 | 0.31 | 16.49 | 9.45 | 0.222 | 9 | 0.17 | 0.065 | −0.01 | 0.36 | 46.53 | 14.41 | 0.072 |
| | **Internet-based TD-CBT versus** | Control | – | – | – | – | – | – | – | – | – | – | – | – | – | – | – | – |
| **12 month FU** | **TD-CBT (all treatment formats) versus** | Control | 11 | 0.24 | <0.001 | 0.13 | 0.35 | 33.53 | 13.48 | 0.198 | 11 | 0.22 | <0.001 | 0.12 | 0.32 | 24.83 | 10.52 | 0.396 |
| | | DS-CBT | 4 | 0.13 | 0.247 | −0.09 | 0.36 | 58.91 | 7.48 | 0.058 | 4 | 0.08 | 0.253 | −0.06 | 0.23 | 0 | 0.87 | 0.832 |
| | | TAU | 4 | 0.35 | <0.001 | 0.24 | 0.47 | 0 | 0.30 | 0.960 | 4 | 0.36 | <0.001 | 0.24 | 0.48 | 0 | 0.70 | 0.872 |
| | | Other | 3 | 0.23 | 0.030 | 0.02 | 0.44 | 0 | 0.03 | 0.985 | 3 | 0.21 | 0.044 | 0.01 | 0.42 | 0 | 0.10 | 0.950 |
| | | WL | – | – | – | – | – | – | – | – | – | – | – | – | – | – | – | – |
| | **Individual TD-CBT versus** | Control | 3 | 0.33 | <0.001 | 0.21 | 0.45 | 0 | 0.58 | 0.748 | 3 | 0.34 | <0.001 | 0.20 | 0.47 | 9.71 | 1.67 | 0.434 |
| | **Group-based TD-CBT versus** | Control | 3 | 0.30 | 0.006 | 0.09 | 0.51 | 0 | 0.59 | 0.745 | 3 | 0.27 | 0.012 | 0.06 | 0.48 | 0 | 0.24 | 0.888 |
| | **Internet-based TD-CBT versus** | Control | 5 | 0.17 | 0.098 | −0.03 | 0.38 | 54.63 | 8.93 | 0.063 | 5 | 0.11 | 0.116 | −0.03 | 0.24 | 0 | 1.95 | 0.746 |
| **24 month FU** | **TD-CBT (all treatment formats) versus** | Control | 5 | 0.20 | 0.111 | −0.05 | 0.46 | 80.63 | 16.44 | 0.003 | 5 | 0.14 | 0.092 | −0.02 | 0.31 | 56.46 | 8.63 | 0.071 |
| | | DS-CBT | 4 | 0.20 | 0.259 | −0.14 | 0.54 | 81.90 | 16.15 | 0.001 | 4 | 0.11 | 0.344 | −0.12 | 0.35 | 62.33 | 8.25 | 0.041 |
| | | TAU | – | – | – | – | – | – | – | – | – | – | – | – | – | – | – | – |
| | | Other | – | – | – | – | – | – | – | – | – | – | – | – | – | – | – | – |
| | | WL | – | – | – | – | – | – | – | – | – | – | – | – | – | – | – | – |
| | **Individual TD-CBT versus** | Control | – | – | – | – | – | – | – | – | – | – | – | – | – | – | – | – |
| | **Group-based TD-CBT versus** | Control | – | – | – | – | – | – | – | – | – | – | – | – | – | – | – | – |
| | **Internet-based TD-CBT versus** | Control | 4 | 0.20 | 0.259 | −0.14 | 0.54 | 81.90 | 16.15 | 0.001 | 4 | 0.11 | 0.344 | −0.12 | 0.35 | 62.33 | 8.25 | 0.041 |

FU, follow-up.

and for posttreatment as well as follow-ups. In addition, effect sizes and confidence intervals (CI) comparing TD-CBT to control for all three settings are displayed in the forest plots in Figs. 3 and 4 (posttreatment). Forest plots for the follow-up assessments are included in Supplementary Figs. 2–9.

Across settings, TD-CBT revealed significantly stronger symptom reduction in depression ($g = 0.74$, 95% CI = 0.57–0.92, $P < 0.001$) and anxiety ($g = 0.77$, 95% CI = 0.56–0.97, $P < 0.001$) than controls at posttreatment. TD-CBT showed superiority to waitlist for depression

($g = 1.23$, 95% CI = 0.80–1.66, $P < 0.001$) and anxiety ($g = 1.24$, 95% CI = 0.82–1.67, $P < 0.001$) and to TAU for depression ($g = 0.90$, 95% CI = 0.66–1.14, $P < 0.001$) and anxiety outcomes ($g = 0.98$, 95% CI = 0.63–1.33, $P < 0.001$) with large effects. We found no statistically significant difference between TD-CBT and DS-CBT in alleviating depressive ($g = 0.09$, 95% CI = −0.07–0.25, $P = 0.269$) and anxiety symptoms ($g = 0.09$, 95% CI = −0.01–0.20, $P = 0.091$). The comparison between TD-CBT and DS-CBT was corroborated by conducting more Bayesian analyses. A description of the statistical procedure for the

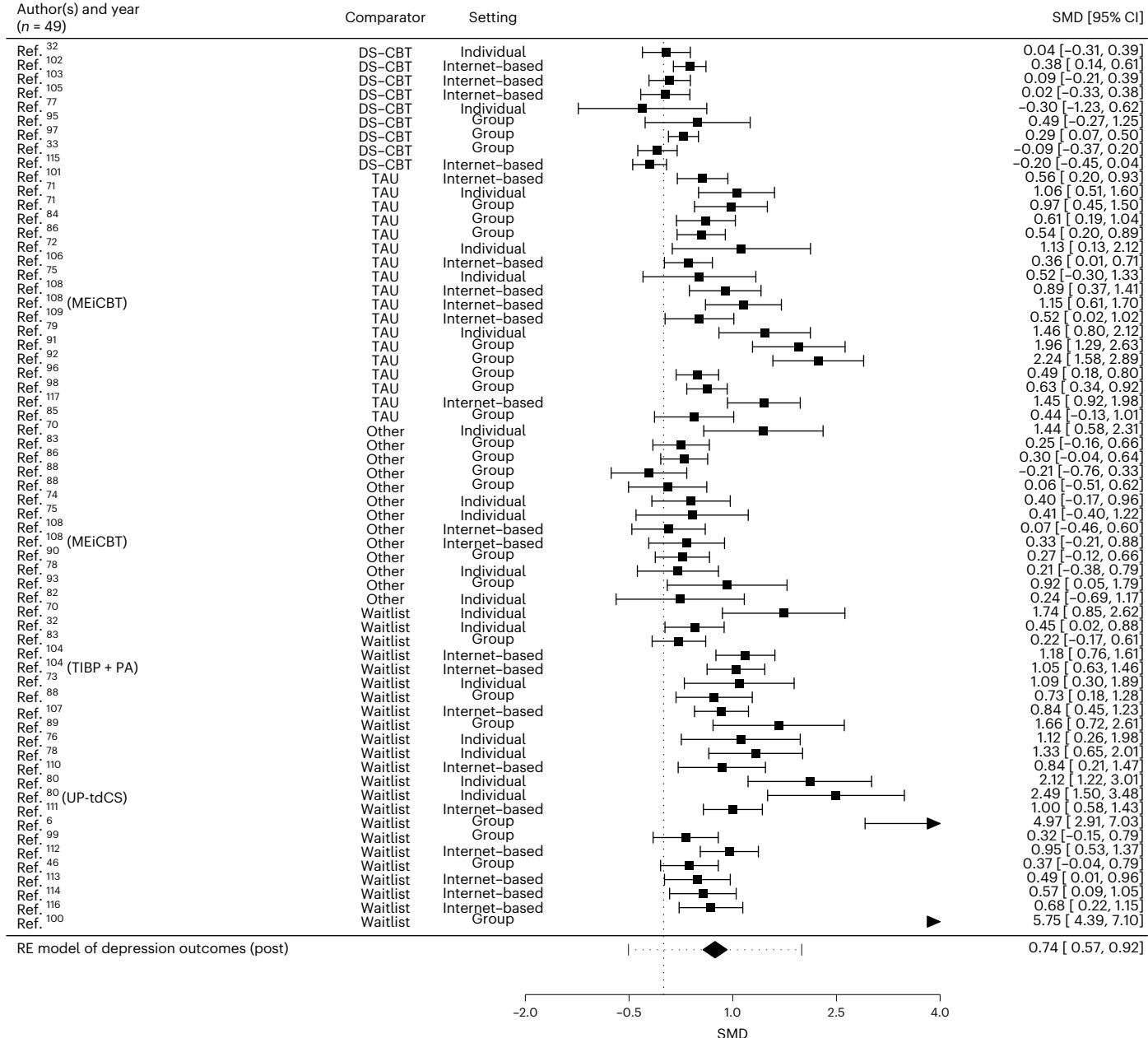

**Fig. 3 | Forest plots of controlled effect sizes (posttreatment) for depression.** Studies are clustered according to the setting in which they investigated TD-CBT. One study[88] compared TD-CBT to ACT and BA. We used a random-effects (RE) model to estimate pooled effects. *n* denotes the number of studies included.

For each study, the black square represents the effect size (standardized mean difference, SMD) and the horizontal bars represent the 95% CI. The overall estimated effect size (Hedges' *g*) is depicted by the diamond with the dotted bars representing its 95% CI.

Bayesian analyses as well as forest plots for the original model and sensitivity analyses can be found in Supplementary Figs. 22–27. Estimated effect sizes confirmed the frequentist findings for depression ($g = 0.09$, 95% CI = −0.12–0.27) and anxiety ($g = 0.09$, 95% CI = −0.04–0.24). In comparison to other active control groups (including bona fide treatments), TD-CBT was more effective for depression ($g = 0.27$, 95% CI = 0.13–0.42, $P < 0.001$) with small effects but not for anxiety ($g = 0.14$, 95% CI = −0.04–0.31, $P = 0.128$). TD-CBT was superior to controls at 3 months follow-up (depression $g = 0.55$, 95% CI = 0.30–0.80, $P < 0.001$; anxiety $g = 0.48$, 95% CI = 0.18–0.79, $P = 0.002$), at 6 months follow-up (depression $g = 0.20$, 95% CI = 0.10–0.30, $P < 0.001$; anxiety $g = 0.23$, 95% CI = 0.11–0.36, $P < 0.001$) and at 12 months follow-up (depression $g = 0.24$, 95% CI = 0.13–0.35, $P < 0.001$; anxiety $g = 0.22$, 95% CI = 0.12–0.32, $P < 0.001$) but not at 24 months follow-up

(depression $g = 0.20$, 95% CI = −0.05–0.46, $P = 0.111$; anxiety $g = 0.14$, 95% CI = −0.02–0.31, $P = 0.092$). Overall, we found high and significant heterogeneity amongst studies (against all controls at posttreatment: $I^2 = 88.29\%$ for depression and $I^2 = 91.72\%$ for anxiety), which remained high after isolating treatment format. Results for sensitivity analyses are provided in Supplementary Tables 5–8. When removing outliers (Supplementary Tables 5 and 6) heterogeneity was reduced with comparable effects.

**Uncontrolled effect sizes.** Uncontrolled effect sizes and their CI as well as measures of heterogeneity ($Q$ statistic and $I^2$) for anxiety and depression outcomes for all three settings, from pre- to postassessment and at follow-ups, are reported in Supplementary Table 4 and Supplementary Figs. 10–19.

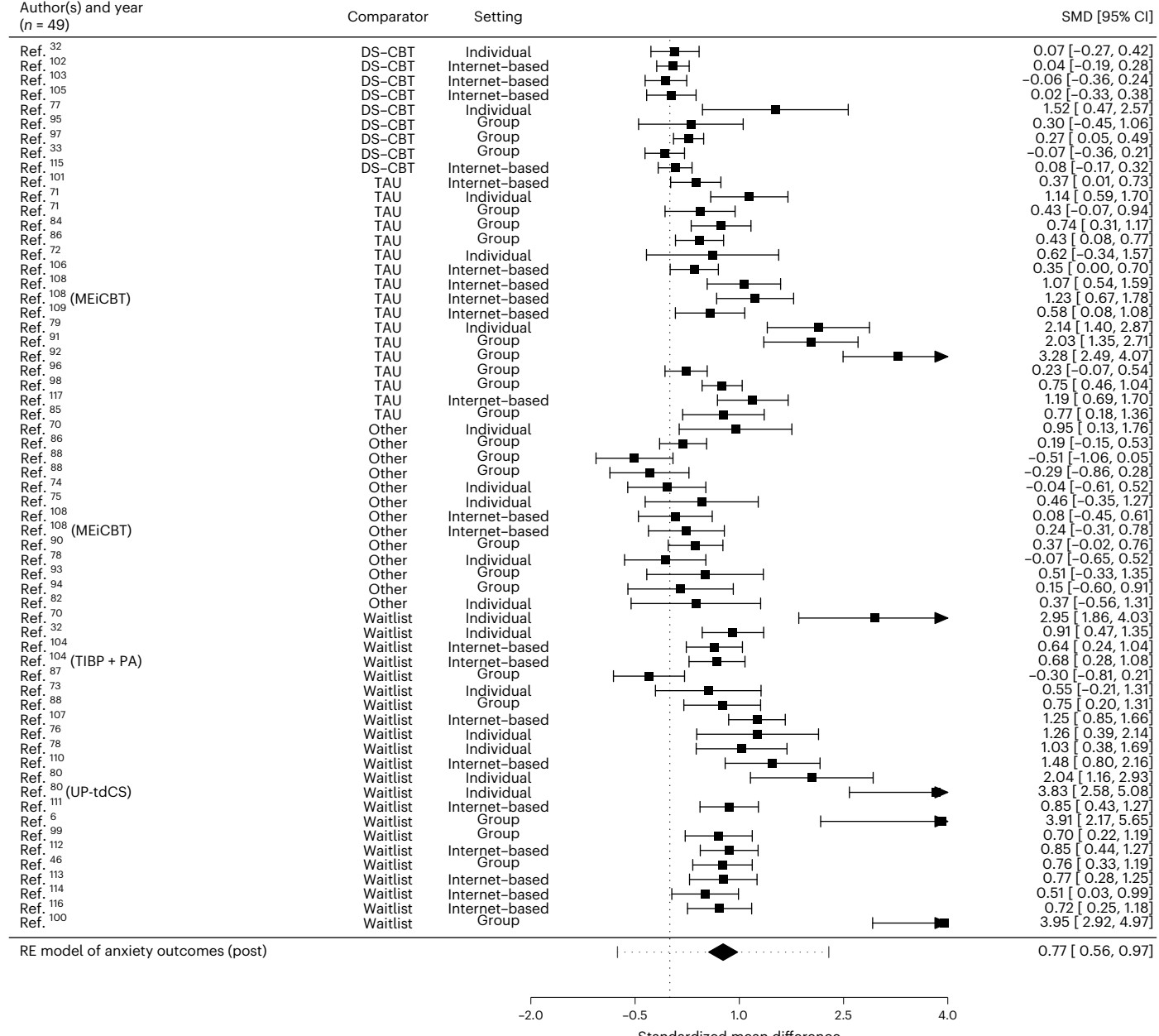

**Fig. 4 | Forest plots of controlled effect sizes (posttreatment) for anxiety.** Studies are clustered according to the setting in which they investigated TD-CBT. One study[88] compared TD-CBT to ACT and BA. We used an RE model to estimate pooled effects. *n* denotes the number of studies included. For each study, the black square represents the effect size SMD and the horizontal bars represent the 95% CI. The overall estimated effect size (Hedges' *g*) is depicted by the diamond with the dotted bars representing its 95% CI.

**Publication bias.** Statistical analyses indicated asymmetry of funnel plots for controlled effects (Kendall's tau = 0.36–0.38, $P < 0.001$; Egger's test $Z = 6.89$–7.45, $P < 0.001$). Funnel plots, with and without trim and fill method, for the controlled effects for anxiety and depression (posttreatment) are provided in Supplementary Figs. 20 and 21.

## Discussion

TD-CBT for emotional disorders attracted increased attention and considerable research activity in recent years. This is reflected in the large number of 56 RCTs with 6,916 patients included in our comprehensive review on individual, group and internet-based formats.

Overall, TD-CBT was effective in both the short and long terms. Most studies compared TD-CBT to waitlist-control conditions and yielded large effect sizes in line with previous benchmarks[30]. Our review

and meta-analysis also included active control groups. We found that TD-CBT produced large effects in comparison to TAU which comprised very heterogeneous setups, ranging from low-key treatments to clinician-tailored personalized interventions (for example, ref. 31). In comparison to other active treatments, such as behavioural activation or CBT for perfectionism, TD-CBT had a stronger impact on depression, with small effects but not on anxiety. Of special interest is how TD-CBT compares to gold-standard DS-CBT (for example, evidence-based manualized individual therapy in one trial[32] or group treatments in another trial[33]). Overall, TD-CBT produced comparable effects to DS-CBT, with no significant differences emerging between both approaches. The comparability of TD-CBT to DS-CBT was investigated in previous meta-analyses with mixed findings: ref. 22 found TD-CBT to produce comparable effects to DS-CBT for anxiety outcomes

but to surpass DS-CBT in its efficacy for depression outcomes. While ref. 15 found TD-CBT to produce significantly greater effects than DS-CBT, their results also suggested that these differences may not be clinically significant. With our meta-analysis including more studies that directly compare TD-CBT and DS-CBT in RCTs, our findings provide further evidence of the comparability of TD-CBT and DS-CBT.

We have also investigated effects beyond the immediate end of treatment. While uncontrolled effects should be interpreted cautiously as they cannot be causally interpreted, we did find that the effects of TD-CBT remained stable over time, based on follow-up assessments at 3, 6 and 12 months. Five studies—four of them stemming from the same research group and investigating internet-based interventions—also included a long-term follow-up up to 24 months. For this long-term follow-up, we found large effects over time for TD-CBT on anxiety and depression outcomes (standardized mean changes ($d_{SMC}$) = 1.47–1.75), with no significant differences between TD-CBT and DS-CBT. While more research by independent groups is warranted, this comparability underlines the potential of TD-CBT and strengthens the argument for a broadly applicable, transdiagnostic approach, with high scalability and reach.

Concerning the different settings we have investigated, we found comparable effects between all three settings, individual, group and internet-based. There was a strong uptake of TD protocols in group and internet-based settings. Applying TD-CBT as a group treatment may be beneficial in health care systems with limited resources. On top of groups saving therapeutic resources, TD-CBT groups specifically may be a more feasible approach to delivering evidence-based care than offering disorder-specific groups. The comparable effects of the individual and internet-based setting strengthen the evidence for the comparability of both settings[12,34]. Delivering TD protocols online or in conjunction with in-person sessions ('blended care') may even boost the potential of TD-CBT to reduce the treatment gap for those in need: TD internet-based interventions are not only highly effective, they also address barriers to treatment access and can reach underserved communities, such as those living in geographically remote areas (for example, refs. 35,36) or those with limited mobility, for example, due to chronic physical conditions[37].

Our study is not without limitations. We included anxiety, obsessive compulsive, depressive as well as adjustment disorders as primary diagnoses in our review, with most studies investigating TD-CBT for GAD, SAD and MDD. We can neither draw conclusions about the efficacy of TD-CBT for individual diagnoses nor judge its efficacy for other diagnoses which, depending on the definition, are also counted among the emotional disorders (for example, somatic symptom disorders, post-traumatic stress disorder or borderline personality disorder). TD-CBT was investigated in different continents and countries, speaking to its dissemination potential. However, investigations from South America and Africa were under-represented.

The risk of bias assessment revealed possible sources of bias especially in terms of blinding of assessors, patients and therapists—which can be expected given our focus on self-report measures and psychotherapy trials. However, we also found concerns in terms of selective reporting which highlighted that more open science practices in psychotherapy research are warranted, from preregistered analyses plans to open data sharing. This would facilitate replications by independent research groups which are needed to explore the generalizability of our findings and preclude allegiance effects which some of our included studies may be at risk of[38–40]. The implementation of such practices may also help to counteract publication bias. We found exceptionally high heterogeneity of effects. Overall heterogeneity decreased when taking treatment format into account and removing outliers. Future research should investigate whether other clinical or methodological factors such as mechanisms targeted in the TD-CBT protocol, treatment dose, patient or study characteristics might have an impact. An individual participant data meta-analysis would be a key next step in this regard. It also remains unclear if there are any contraindications for TD-CBT, since symptom deterioration, comorbidity and dropout were not systematically examined. Moreover, the clinical relevance of symptom improvement is yet to be investigated and outcome measures beyond symptoms of depression and anxiety, such as quality of life or level of functioning, should be explored. We chose to exclusively study adult populations for our investigation of TD-CBT due to differences in developmental adaptations of treatments, classifications of emotional disorders, outcome measures and treatment efficacy between child/adolescent and adult populations. We focused our review on unified 'broadband' TD-CBT that aims at changing mechanisms shared between disorders. With the surge of research on personalized interventions[41], it may be a fruitful next step to investigate the merit of personalizing unified TD-CBT interventions as well. TD-CBT promises to facilitate training and clinical decision-making, rendering training and treatment less costly. A first study investigated the cost-effectiveness of TD-CBT and found that it may be a cost-effective alternative to TAU[42,43]. However, more research on whether the proposed advantages, for example, in terms of training times and cost-effectiveness, generally hold true is needed.

Our analyses provide evidence that TD-CBT in face-to-face individual, group and internet-based formats is efficacious in reducing symptoms of anxiety and depression. Evidence from trials on internet-based TD-CBT revealed large and stable long-term effects. Taken together, these findings further strengthen the transdiagnostic approach to the treatment of emotional disorders across settings.

## Methods

This study is based exclusively on published literature, therefore no ethics approval was required. In conducting and reporting this review and meta-analysis, we followed the Cochrane Handbook for systematic reviews[44] and the updated PRISMA[45]. The protocol was registered with PROSPERO on 27 September 2019 (registration no. CRD42019141512).

### Search strategy

A systematic literature search was conducted on PubMed and MEDLINE, PsycINFO, Google Scholar, medRxiv (including bioRxiv) and OSF Preprints up to 16 June 2023. Different to what we preregistered, our search covered preprint servers to consider also the most recent findings. We built a search string by combining the concepts 'transdiagnostic', 'CBT', 'emotional disorder' and 'RCT' using the AND Boolean operator. Each concept included terms connected with the OR Boolean operator. The concept 'CBT' also covered terms describing the treatment setting (for example, 'internet-based intervention'). We searched for relevant medical subject headings (MeSH), used by the United States National Library of Medicine to index articles in PubMed and MEDLINE (for example, 'cognitive behavioural therapy', 'anxiety', 'depression'). In addition, we included terms commonly used in the relevant literature (for example, 'unified'). The resulting string was then slightly adapted according to the search options of the different databases (see Supplementary Table 2 for the complete search strings). We included additional studies if they were identified by reference lists and met our inclusion criteria. We used Zotero (v.6.0.23) and Google Sheets.

### Inclusion criteria

We included studies published between January 2000 up until June 2023.

**Population.** Studies were included if the treatment was delivered to treatment-seeking adults. Deviating from our preregistration, we did not apply an upper age limit of 65 years, if the study was not solely targeted at older adults and the mean age of the study population was comparable to other studies with adult populations. We opted for this change to provide a more comprehensive review. Participants had at least one clinician-established diagnosis of an emotional disorder. We included SAD, panic disorder, agoraphobia, GAD, obsessive

compulsive disorder, unipolar depressive disorders and adjustment disorders as treatment targets.

**Interventions.** We selected studies that investigated TD-CBT in an individual, group or internet-based setting (with or without clinician guidance). This included established unified comprehensive TD protocols that were specifically developed to target underlying processes or comorbidity such as the UP[5], false behaviour elimination therapy[46], emotion regulation therapy[47], affect regulation training[48] or transdiagnostic behaviour therapy[7]. We also included protocols that contained CBT components that modified dysfunctional cognitions and behavioural patterns across diagnostic groups, for example, cognitive restructuring. Following this definition, we included 'common elements approaches'[49] if they were presented in a UP, that is, a combination of effective components across disorders. As our focus was not on third wave or experiential approaches within CBT, we excluded standalone mindfulness-based treatment approaches[50,51], metacognitive therapy/training[52] and acceptance and commitment therapy[53,54]. We also excluded protocols targeting transdiagnostic phenomena that cannot be considered shared mechanisms between disorders, such as protocols focusing on self-worth or loneliness.

**Comparison groups.** We included studies that compared TD-CBT to a control group, including (1) waitlist-control condition (that is, delayed treatment), (2) TAU, (3) DS-CBT and (4) other active psychological interventions. TAU included all treatments that the original study defined as 'usual care', 'standard care' or 'care as usual'[55]. Other active psychological interventions included interventions that are based on a psychological rationale but are neither considered TAU nor diagnosis-specific treatments, for example, behavioural activation.

**Outcomes.** Included studies applied a continuous self-report measure of anxiety and/or depression severity at pre- and posttreatment and (if available) at follow-up.

**Study design.** RCTs were included.

### Exclusion of studies
We excluded studies if (1) the treatment was not based on CBT principles, such as psychodynamic interventions and process-experiential principles and (2) they investigated a modularized or tailored treatment, as we did not consider this in line with the concept of unified TD-CBT. Supplementary Table 3 provides an overview of reasons for exclusions for all excluded studies that were full-text screened.

### Study selection and data extraction
Two reviewers independently screened search results based on title and abstract, evaluated potentially eligible publications through full-text read, selected studies matching the inclusion criteria and extracted data for the meta-analysis. Selection results were compared and any disagreements about eligibility were resolved through discussion and in consultation with the project leaders. Interrater-agreement was reached for 95% of the reviewed studies. If not reported in the publication, data were requested directly from study authors. We contacted 35 authors and sent up to two follow-up emails in case of no response, 69% of the authors sent us requested data. We extracted means and standard deviations of self-reported anxiety and/or depression at all available time points corresponding to pre- and posttreatment as well as follow-up. We grouped follow-up time points on the basis of the most frequent reassessments in the included studies. Most of the studies reassessed participants at exactly 3 months ($n = 22$ studies), 6 months ($n = 16$ studies), 12 months ($n = 10$ studies) or 24 months ($n = 4$ studies). Only one study had a shorter follow-up than 3 months, two studies had a follow-up between 3 and 6 months, four studies between 6 and 12 months and one study between 12 and 24 months. For the few studies

with follow-ups falling in between those four measurement points, we allocated the data to the time point to which they were closest.

As many studies reported more than one outcome measure for anxiety or depression, we used the primary outcome measure defined by the study authors or, if this was not available, the measure most commonly used across studies in our final sample. Other variables extracted were control group (waitlist/TAU/DS-CBT/other) and treatment setting (individual/group/internet-based). Studies were grouped for synthesis by type of control group and treatment setting.

### Statistical analyses
All analyses were conducted in R (v.4.3.1), using the metafor[56] (v.4.2-0), meta[57] (v.6.5-0) and dmetar packages[58] (v.0.1.0).

We calculated controlled effect sizes for the difference between the transdiagnostic treatment and the control conditions in main outcomes (depression and anxiety) at posttreatment (relative efficacy), using the bias-corrected Hedges' g and the 95% CI[59]. These were calculated by subtracting the mean posttreatment score of the transdiagnostic condition from the mean score of the control condition, divided by the pooled standard deviation of both conditions. Values of 0.2, 0.5 and 0.8 of Hedges' g represent a small, moderate and large effect size, respectively[60].

Building on previous work[14,22], we expected considerable variability and thus used a random-effects model[61] to account for heterogeneity of included studies[62]. We tested heterogeneity of effect sizes with the Q statistic, the $I^2$ statistic and by visual inspection of forest plots. A P value of the Q statistic below 0.05 indicates heterogeneity[63]. $I^2$ ranges from 0 to 100%, with 25% representing low, 50% moderate and 75% high heterogeneity[64]. We addressed heterogeneous effect sizes by conducting subgroup analyses for the three different treatment formats (individual, group or internet-based), if at least three studies per subgroup were available. Additionally, we investigated whether excluding outliers impacted effect sizes and heterogeneity. In line with previous meta-analyses, outliers were defined as studies whose 95% CI did not overlap with the 95% CI of the overall effect size[20].

We calculated uncontrolled effect sizes from pre- to posttreatment (absolute efficacy) for main outcomes (depression and anxiety) and from pretreatment to follow-up assessment. If reported, we used the intention-to-treat data from the studies for these analyses. As recommended by ref. 56, we estimated the uncontrolled effect sizes using $d_{SMC}$ and their respective 95% CI. Raw score standardization with heteroscedastic population variances at baseline (pretreatment) and posttreatment/follow-up were applied for more reliable estimates[65,66]. The effect sizes $d_{SMC}$ were determined using the means, standard deviations (s.d.) at each time point and the retest correlation between these time points. Values of 0.2, 0.5 and 0.8 for $d_{SMC}$ represent a small, moderate and large effect size, respectively[60]. If the correlation was not available from the studies included, retest correlations were calculated from the original study data. If not available, a default value of 0.5 was set[61]. In addition, we performed sensitivity analyses using 0.3 and 0.7 as retest correlations.

We assessed publication bias by inspecting the funnel plot on the depression and anxiety outcome measures as well as calculating rank correlations and Egger's tests. Additionally, we applied the Trim and Fill procedure[67].

### Study quality assessment
As an updated version of the Cochrane risk-of-bias tool had become available since we registered this review on PROSPERO, deviating from our preregistration, we evaluated the risk of bias of the studies by using the revised Cochrane risk-of-bias tool (RoB 2.0)[68]. We assessed the risk as 'low', 'some concerns' or 'high' in the following five domains: (1) bias of the randomization process; (2) bias of deviations from intended interventions; (3) bias of missing outcome data; (4) bias in measurement of the outcome; and (5) bias in selection of the reported results.

Each domain is made up of several criteria. For example, the first question in domain (1) asks 'Was the allocation sequence random?' which, after consulting the respective manuscript, is answered as 'yes', 'probably yes', 'probably no', 'no' or 'no information'. The RoB 2.0 provides examples and decision trees that clearly specify that certain combinations of ratings across questions within a domain result in the risk of bias of that domain being rated as 'low', 'some concerns' or 'high'. In domain (1), a 'high' risk of bias would be noted if differences between intervention groups were evident at baseline, suggesting a problem with the randomization—regardless of whether no risk of bias was indicated in the evaluation of all other criteria. Two reviewers independently rated each study for bias. Final assessments were cross-checked and disagreements were resolved through discussions between the reviewers. We created the visualization of the risk of bias assessment with the shiny app robvis[69].

### Reporting summary

Further information on research design is available in the Nature Portfolio Reporting Summary linked to this article.

## Data availability

The data that support the findings of this study, along with data collection templates, are publicly available at the Open Science Framework and can be accessed at https://osf.io/ta4fg/.

## Code availability

Custom analysis code that supports the findings of this study is publicly available at the Open Science Framework and can be accessed at https://osf.io/ta4fg/.

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

## Acknowledgements

This study was funded by a Swiss National Science Foundation grant no. 169827 (B.K.) and Freie Universitaet Berlin/University of Zurich joint seed funding, FSP-2019-501 (B.K. and B.R). The funders had no role in study design, data collection and analysis, decision to publish or preparation of the manuscript. We thank L. Bakirtas and A. Olsson for their help with data extraction and all authors who provided additional data or information on their papers, supporting this review and meta-analysis.

## Author contributions

A.S. and B.K. conceived this work. A.S., J.B., B.R., C.F. and B.K. developed the methodology. C.S., L.M., A.S. and M.W. produced the software. C.S., L.M., A.S. and M.W. undertook the formal analysis. C.S., L.M. and A.S. undertook the investigation. B.K. obtained resources. C.S., L.M. and A.S. undertook data curation. C.S. created the visualizations. J.B., B.R., C.F. and B.K. supervised the project. B.K. was responsible for project administration. B.K. and B.R. obtained funding. C.S., L.M. and A.S. wrote the original draft of the paper. C.S., L.M., A.S., M.W., A.M., C.P., D.R., J.B., B.R., C.F. and B.K. were involved in reviewing and editing.

## Competing interests

The authors declare no competing interests.

## Additional information

**Correspondence and requests for materials** should be addressed to Carmen Schaeuffele or Laura E. Meine.

# Reporting Summary

## Statistics

For all statistical analyses, confirm that the following items are present in the figure legend, table legend, main text, or Methods section.

| n/a | Confirmed | |
|---|---|---|
| ☐ | ☒ | The exact sample size (*n*) for each experimental group/condition, given as a discrete number and unit of measurement |
| ☒ | ☐ | A statement on whether measurements were taken from distinct samples or whether the same sample was measured repeatedly |
| ☐ | ☒ | The statistical test(s) used AND whether they are one- or two-sided *Only common tests should be described solely by name; describe more complex techniques in the Methods section.* |
| ☒ | ☐ | A description of all covariates tested |
| ☐ | ☒ | A description of any assumptions or corrections, such as tests of normality and adjustment for multiple comparisons |
| ☐ | ☒ | A full description of the statistical parameters including central tendency (e.g. means) or other basic estimates (e.g. regression coefficient) AND variation (e.g. standard deviation) or associated estimates of uncertainty (e.g. confidence intervals) |
| ☐ | ☒ | For null hypothesis testing, the test statistic (e.g. *F*, *t*, *r*) with confidence intervals, effect sizes, degrees of freedom and *P* value noted *Give P values as exact values whenever suitable.* |
| ☒ | ☐ | For Bayesian analysis, information on the choice of priors and Markov chain Monte Carlo settings |
| ☒ | ☐ | For hierarchical and complex designs, identification of the appropriate level for tests and full reporting of outcomes |
| ☐ | ☒ | Estimates of effect sizes (e.g. Cohen's *d*, Pearson's *r*), indicating how they were calculated |

*Our web collection on statistics for biologists contains articles on many of the points above.*

## Software and code

Policy information about availability of computer code

| Data collection | Zotero (Version 6.0.23), Google Sheets |
|---|---|
| Data analysis | All analyses were conducted in R (version 4.3.1), using the metafor (Viechtbauer, 2010; version 4.2-0), meta (Balduzzi et al., 2019; version 6.5-0), and dmetar package (Harrer et al., 2019; version 0.1.0). Risk of bias was evaluated with the Cochrane risk-of-bias tool (RoB 2.0). All code is available on the Open Science Framework (OSF) at https://osf.io/ta4fg. |

For manuscripts utilizing custom algorithms or software that are central to the research but not yet described in published literature, software must be made available to editors and reviewers. We strongly encourage code deposition in a community repository (e.g. GitHub). See the Nature Portfolio guidelines for submitting code & software for further information.

## Data

Policy information about availability of data

All manuscripts must include a data availability statement. This statement should provide the following information, where applicable:
- Accession codes, unique identifiers, or web links for publicly available datasets
- A description of any restrictions on data availability
- For clinical datasets or third party data, please ensure that the statement adheres to our policy

All data for this project, along with relevant code and codebooks, are publicly available on the OSF at https://osf.io/ta4fg.

# Research involving human participants, their data, or biological material

Policy information about studies with underline{human participants or human data}. See also policy information about underline{sex, gender (identity/presentation), and sexual orientation} and underline{race, ethnicity and racism}.

**Reporting on sex and gender**
*Use the terms sex (biological attribute) and gender (shaped by social and cultural circumstances) carefully in order to avoid confusing both terms. Indicate if findings apply to only one sex or gender; describe whether sex and gender were considered in study design; whether sex and/or gender was determined based on self-reporting or assigned and methods used.*
*Provide in the source data disaggregated sex and gender data, where this information has been collected, and if consent has been obtained for sharing of individual-level data; provide overall numbers in this Reporting Summary. Please state if this information has not been collected.*
*Report sex- and gender-based analyses where performed, justify reasons for lack of sex- and gender-based analysis.*

**Reporting on race, ethnicity, or other socially relevant groupings**
*Please specify the socially constructed or socially relevant categorization variable(s) used in your manuscript and explain why they were used. Please note that such variables should not be used as proxies for other socially constructed/relevant variables (for example, race/ethnicity should not be used as a proxy for socioeconomic status).*
*Provide clear definitions of the relevant terms used, how they were provided (by the participants/respondents, the researchers, or third parties), and the method(s) used to classify people into the different categories (e.g. self-report, census or administrative data, social media data, etc.)*
*Please provide details about how you controlled for confounding variables in your analyses.*

**Population characteristics**
*Describe the covariate-relevant population characteristics of the human research participants (e.g. age, genotypic information, past and current diagnosis and treatment categories). If you filled out the behavioural & social sciences study design questions and have nothing to add here, write "See above."*

**Recruitment**
*Describe how participants were recruited. Outline any potential self-selection bias or other biases that may be present and how these are likely to impact results.*

**Ethics oversight**
*Identify the organization(s) that approved the study protocol.*

Note that full information on the approval of the study protocol must also be provided in the manuscript.

# Field-specific reporting

Please select the one below that is the best fit for your research. If you are not sure, read the appropriate sections before making your selection.

☐ Life sciences   ☒ Behavioural & social sciences   ☐ Ecological, evolutionary & environmental sciences

For a reference copy of the document with all sections, see nature.com/documents/nr-reporting-summary-flat.pdf

# Behavioural & social sciences study design

All studies must disclose on these points even when the disclosure is negative.

**Study description**
Systematic review and meta-analysis with quantitative analysis of sub-groups

**Research sample**
Randomised controlled trials evaluating transdiagnostic, CBT-based psychotherapy treatments (TD-CBT) in adult patients diagnosed with an emotional disorder. We included studies published from January 2000 that investigated TD-CBT in individual, group, or Internet-based setting compared to a wait-list control condition, treatment-as-usual, disorder-specific treatment or another active psychological intervention. Studies needed to include a self-report measure of anxiety and/or depression severity at pre- and post-treatment. The dataset compiled from extracted data of included studies is publicly available on the OSF.

**Sampling strategy**
We formulated clear inclusion and exclusion criteria and then conducted a systematic literature search on PubMed and MEDLINE, PsycINFO, Google Scholar, medRxiv (incl. bioRxiv), and OSF Preprints up to 12/01/2023. All studies that met the inclusion criteria and did not fulfill any exclusion criterion were included for analyses.

**Data collection**
Articles found through the above mentioned systematic search that matched our inclusion criteria were collected in Zotero. Data for the meta-analysis was extracted by two independent reviewers and saved in a Google sheet (see available data). As this was a systematic review and meta-analysis, the researchers were not blinded.

**Timing**
Studies published between January 2000 and 12/01/2023 were included.

**Data exclusions**
For the meta-analytic calculations, we excluded one study because the data were not available and another two studies because no self-report of anxiety or depression was available.

**Non-participation**
129 studies were full-text screened but met at least one exclusion criterion and were therefore not considered for analyses. Reasons for exclusions were: secondary analysis (n = 18), no clinician-established diagnosis (n = 35), different primary diagnosis (n = 4), not based on transdiagnostic CBT principles (n = 19), tailored treatment (n = 8), anxiety/depression outcome not available (n = 2), no RCT (n = 19), study protocol (n = 8), meta-analysis (n = 2), factual text (n = 2), other (n = 12).

| Randomization | Only RCTs were included in this review and meta-analysis. |
|---|---|

# Reporting for specific materials, systems and methods

We require information from authors about some types of materials, experimental systems and methods used in many studies. Here, indicate whether each material, system or method listed is relevant to your study. If you are not sure if a list item applies to your research, read the appropriate section before selecting a response.

## Materials & experimental systems

| n/a | Involved in the study |
|---|---|
| ☒ ☐ | Antibodies |
| ☒ ☐ | Eukaryotic cell lines |
| ☒ ☐ | Palaeontology and archaeology |
| ☒ ☐ | Animals and other organisms |
| ☒ ☐ | Clinical data |
| ☒ ☐ | Dual use research of concern |
| ☒ ☐ | Plants |

## Methods

| n/a | Involved in the study |
|---|---|
| ☒ ☐ | ChIP-seq |
| ☒ ☐ | Flow cytometry |
| ☒ ☐ | MRI-based neuroimaging |

