## [Peer Review File · Nature Human Behaviour]

Peer Review Information

Journal: Nature Human Behaviour

Manuscript Title: A systematic review and meta-analysis of transdiagnostic cognitive behavioral therapies for emotional disorders

Corresponding author name(s): Carmen Schaeuffele and Laura Meine

Reviewer Comments & Decisions:

Decision Letter, initial version:
--

16th June 2023

Dear Dr Meine,

Thank you once again for your manuscript, entitled "Short- and Long-term Efficacy of Transdiagnostic Cognitive Behavioral Therapies for Emotional Disorders: Systematic Review and Meta-Analysis," and for your patience during the peer review process.

Your manuscript has now been evaluated by 3 reviewers, whose comments are included at the end of this letter. Although the reviewers find your work to be of interest, they also raise some important concerns. We are very interested in the possibility of publishing your study in Nature Human Behaviour, but would like to consider your response to these concerns in the form of a revised manuscript before we make a decision on publication.

To guide the scope of the revisions, the editors discuss the referee reports in detail within the team, including with the chief editor, with a view to (1) identifying key priorities that should be addressed in revision and (2) overruling referee requests that are deemed beyond the scope of the current study. We hope that you will find the prioritised set of referee points to be useful when revising your study. Please do not hesitate to get in touch if you would like to discuss these issues further.

1. Reviewer 1 and 2 raise concerns about the definition of transdiagnostic CBT approaches and the exclusion of modularized or tailored treatments. Please provide a clear definition of transdiagnostic CBT approaches, and explain how modularized or tailored treatments differ from those. In addition, we ask that you either explicitly motivate the exclusions of modularized or tailored treatments, or include them in your analysis.

2. Reviewer 3 raises concerns about the focus on transdiagnostic CBT approaches for adults only, and the grouping of time-points. Please carefully address these concerns, and clearly motivate all

methodological choices, as well as conduct additional sensitivity analyses as requested by Reviewer 3.

3. Please update your search to ensure that it is up-to-date.

4. Reviewer 3 notes that pre-post effect sizes are not informative and could be misleading (given that they are based on the assumption that nothing other than the treatment has changed). We ask that you follow the reviewer's recommendation to remove these and instead rely on between group effect sizes.

5. Please address all remaining concerns, including the reviewer requests for greater methodological detail, especially with respect to risk of bias, the indicators for which should follow standard practice.

In sum, we invite you to revise your manuscript taking into account all reviewer and editor comments. We are committed to providing a fair and constructive peer-review process. Do not hesitate to contact us if there are specific requests from the reviewers that you believe are technically impossible or unlikely to yield a meaningful outcome.

We hope to receive your revised manuscript within two months. I would be grateful if you could contact us as soon as possible if you foresee difficulties with meeting this target resubmission date.

- Include a "Response to the editors and reviewers" document detailing, point-by-point, how you addressed each editor and referee comment. If no action was taken to address a point, you must provide a compelling argument. When formatting this document, please respond to each reviewer comment individually, including the full text of the reviewer comment verbatim followed by your response to the individual point. This response will be used by the editors to evaluate your revision and sent back to the reviewers along with the revised manuscript.
- Highlight all changes made to your manuscript or provide us with a version that tracks changes.

[REDACTED]

We look forward to seeing the revised manuscript and thank you for the opportunity to review your work. Please do not hesitate to contact me if you have any questions or would like to discuss these

revisions further.

Sincerely,

Samantha Antusch

Samantha Antusch, PhD
Senior Editor
Nature Human Behaviour

Reviewer expertise:

Reviewer #1: transdiagnostic approaches for treating anxiety and depression

Reviewer #2: transdiagnostic approaches for treating anxiety and depression

Reviewer #3: meta-analysis, clinical psychology

REVIEWER COMMENTS:

Reviewer #1:

Remarks to the Author:

This systematic review and meta-analysis is extremely timely and relevant, given the many controlled trials conducted on transdiagnostic therapies in the last five to ten years. Overall, the meta-analysis is very well justified and its design, execution, analysis and presentation are clearly presented and accurately reported. In general the interpretation and scientific context of the findings are also appropriate. There are however three main issues that I believe need addressing:

1) The authors say they excluded "modularized or tailored treatment, as we did not consider this in line with the concept of unified TD-CBT". I could not find a justification for this decision in the Introduction, but I think, regardless, it needs full elaboration, definitions and scientific and/or practical justification in the Methods section. One major concern I have is how the authors could be certain of whether an intervention was or was not provided in a 'modularised' and/or 'tailored' manner. I find it hard to believe that all studies would contain sufficient information regarding their protocol in order to be certain of this either way. Also, I cannot see any reason, in principle, why modularising or tailoring an intervention would make it less a unified TD approach, unless, specifically, the modularisation or tailoring was carried out on the basis of a mental health diagnosis or divide by diagnostic category. If the modules were different steps or processes that are all transdiagnostic, or if the 'tailoring' was to the individual, regardless of their diagnosis, then I would not perceive this to be a deviation from the transdiagnostic principles.

2) There is no mention of this article:

Andersen, P., Toner, P., Bland, M., & McMillan, D. (2016). Effectiveness of transdiagnostic cognitive behaviour therapy for anxiety and depression in adults: a systematic review and meta-analysis.

Behavioural and cognitive psychotherapy, 44(6), 673-690.

I think that the fact that it is 7 years old could be used as a rationale for the submitted article, but I would also suggest providing any other reasons.

3) I feel that the statement that TD-CBT is effective 'in the long term' is glossed over in the abstract and discussion. There is sufficient detail in the Tables, but not in the Methods section. The authors need to do justice to the findings at specific periods post-treatment and make specific interpretations of these findings; this includes the abstract.

Overall, an excellent article, worthwhile of publication in this journal, and its imperfections can be addressed.

Reviewer #2:

Remarks to the Author:

This systematic review and meta-analysis reported on the short and long term efficacy of transdiagnostic cognitive behavioral therapies for emotional disorders. There is much to like about this manuscript, it's well-written, follows PRISMA guidelines, and was pre-registered (with departures from preregistration acknowledged in the manuscript). Limitations are also well considered, and the methods are solid. However, I think clarity is needed in the following ways:

The authors' definition of transdiagnostic CBT is unclear. More specifically the boundaries for what is and what needs greater justification and clarity. For example, the authors include Gonzalez-Robles et al.'s work using a transdiagnostic internet-based emotion regulation protocol but do not include Mennin & Fresco's work on emotion regulation therapy. There are also trials included that seem to include only a single disorder (e.g., Hvenegaard et al., 2019) which seems at odds with transdiagnostic treatment and there is only one trial included on repetitive negative thinking although several other RCTs have been published.

Greater clarity is also needed regarding the rule-out of modularized treatment on pg 9 as the treatment most represented in this meta-analysis is the Unified Protocol (UP) which is a modular treatment. That is, the UP is personalized in some ways as the number of sessions spent in each module is tailored to the individual patient (see Ellard et al., 2010).

A small point. It would be helpful to have some descriptives about the length of treatment (e.g., were any of them designed as brief treatments, were there any outside of the typical 12-18 sessions).

Reviewer #3:

Remarks to the Author:

- Can the authors state whether there were any deviations to the pre-registered protocol? If there weren't, it would still be necessary to state that they adhered to this.
- The authors may wish to consider updating their search as it has been six months since the last search was conducted. It would be necessary to ensure that no new studies are missed during this time.
- Please provide key examples of search terms used in the manuscript, rather than providing the whole string in the supplementary materials.

- Why was the population group limited to adults? Why not include children/adolescents and then conduct sensitivity/subgroup analyses to see if effects vary as a function of age? The authors would need to present some compelling evidence their exclusion.
- How many authors were contacted for data not reported? This information would be worthwhile to include?
- Was there any rationale for grouping the follow-up time-points in this fashion? Providing some justification would be necessary, otherwise perhaps consider conducting sensitivity analyses ensuring that results do not change due to the cut-off decisions made.
- Calculation of pre-post effect sizes is strongly discouraged due the inability to know which portion of the effect results from naturally occurring changes and which from the intervention (see doi:10.1017/S2045796016000809). The authors are discouraged from reporting these and instead focus on between group effect sizes. This is a possibility given that RCTs were only included.
- Please describe the risk of bias indicators in more detail, including what they mean, how they were scored etc. what was the reliability among raters when evaluating risk of bias?
- Please remove all of the citations on Page 13 and instead place them in a table with the study characteristics. It is too difficult to follow at this point.
- It would be necessary to provide numbers pertaining to risk of bias indicators. In other words, how many met quality criteria 1, 2...5? What percentage of studies met all 5 criteria?
- Pre-post effect sizes are likely inflated and tell us very little. The authors should consider removing these.
- How did the authors detect outliers? What was the criteria for determining an outlier?

Author Rebuttal to Initial comments

Reviewer #1

1) The authors say they excluded "modularized or tailored treatment, as we did not consider this in line with the concept of unified TD-CBT". I could not find a justification for this decision in the Introduction, but I think, regardless, it needs full elaboration, definitions and scientific and/or practical justification in the Methods section. One major concern I have is how the authors could be certain of whether an intervention was or was not provided in a 'modularised' and/or 'tailored' manner. I find it hard to believe that all studies would contain sufficient information regarding their protocol in order to be certain of this either way. Also, I cannot see any reason, in principle, why modularising or tailoring an intervention would make it less a unified TD approach, unless, specifically, the modularisation or tailoring was carried out on the basis of a mental health diagnosis or divide by diagnostic category. If the modules were different steps or processes that are all transdiagnostic, or if the 'tailoring' was to the individual, regardless of their diagnosis, then I would not perceive this to be a deviation from the transdiagnostic principles.

see response to editor comment #1

2) There is no mention of this article:
Andersen, P., Toner, P., Bland, M., & McMillan, D. (2016). Effectiveness of transdiagnostic

cognitive behaviour therapy for anxiety and depression in adults: a systematic review and meta-analysis. Behavioural and cognitive psychotherapy, 44(6), 673-690.

I think that the fact that it is 7 years old could be used as a rationale for the submitted article, but I would also suggest providing any other reasons.

Thank you for pointing this out. We have added the review and meta-analysis by Andersen and colleagues to our introduction (p. 6-7, l. 124-128). As the reviewer suggests, since they concluded their search in 2013 and only included 4 RCTs, we feel confident about the added value of our meta-analysis. We also added a meta-analysis on transdiagnostic group treatments that was published since we submitted our manuscript (Joaquim et al., 2023).

3) I feel that the statement that TD-CBT is effective 'in the long term' is glossed over in the abstract and discussion. There is sufficient detail in the Tables, but not in the Methods section. The authors need to do justice to the findings at specific periods post-treatment and make specific interpretations of these findings; this includes the abstract.

We agree that we kept the summary and discussion of long-term effects rather brief in the previous submission and now report these findings in more detail. Please note that the reporting of the longer term effects is presented slightly differently, following the suggestion by reviewer 3 to ungroup follow-up time points and focus on the controlled, rather than the uncontrolled, effects.

Abstract:

TD-CBT was superior to controls at 3-month follow-up (depression: $g = 0.56$, 95%CI = 0.30 - 0.83, $p < 0.001$; anxiety: $g = 0.48$, 95%CI = 0.16 - 0.80, $p = 0.004$), at 6-month follow-up (depression: $g = 0.18$, 95%CI = 0.07 - 0.29, $p = 0.001$; anxiety: $g = 0.2$, 95%CI = 0.09 - 0.31, $p < 0.001$), and at 12-month follow-up (depression: $g = 0.24$, 95%CI = 0.13 - 0.35, $p < 0.001$; anxiety: $g = 0.22$, 95%CI = 0.12 - 0.32, $p < 0.001$) but not at 24-months follow-up (depression: $g = 0.20$, 95%CI = -0.05 - 0.46, $p = 0.111$; anxiety: $g = 0.14$, 95%CI = -0.02 - 0.31, $p = 0.092$).

Discussion (p. 13, l. 271-281):

We have also investigated effects beyond the immediate end of treatment. While uncontrolled effects should be interpreted cautiously as they cannot be causally interpreted, we did find that the effects of TD-CBT remained stable over time, based on follow-up assessments at 3 months, 6 months, and 12 months. Five studies - four of them stemming from the same research group and investigating Internet-based interventions - also included a long-term follow-up up to 24 months: For this long-term follow-up, we found large effects over time for TD-CBT on anxiety and depression outcomes ($d = 1.47 - 1.75$), with no significant differences between TD-CBT and DS-CBT. While more research by independent groups is warranted, this comparability underlines the potential of TD-CBT and strengthens the argument for a broadly applicable, transdiagnostic approach, with high scalability and reach.

Reviewer #2

1) The authors definition of transdiagnostic CBT is unclear. More specifically the boundaries for what is and what needs greater justification and clarity. For example, the author's include Gonzalez-Robles et. al's work using a transdiagnostic internet-based emotion regulation protocol but do not include Mennin & Fresco's work on emotion regulation therapy. There are also trials included that seemed include only a single disorder (e.g., Hvenegaard et al., 2019) which seems at odds with transdiagnostic treatment and there is only one trial included on repetitive negative thinking although several other RCTs have been published.
Thank you for clarifying. Reviewer 1 and the editorial team also raised the topic of the definition of transdiagnostic interventions. See our combined response to editor comment #1. We have indeed also included studies that investigated only a specific disorder, as long as the intervention applied was transdiagnostic. As the protocol applied in the Hvenegaard et al. (2019) trial investigated a disorder-specific approach for depression, it was excluded.
2) Greater clarity is also needed regarding the rule-out of modularized treatment on pg 9 as the treatment most represented in this meta-analysis is the Unified Protocol (UP) which is a modular treatment. That is, the UP is personalized in some ways as the number of sessions spent in each module is tailored to the individual patient (see Ellard et al., 2010).
see response to editor comment #1
3) A small point. It would be helpful to have some descriptives about the length of treatment (e.g., were any of them designed as brief treatments, were there any outside of the typical 12-18 sessions).
To give readers a better understanding of the different treatment lengths, we have added a column to Table 1 that shows the length of treatment for each included study. We now also report the mean number of sessions and the SD in the narrative summary of studies in the results section (p. 8., l. 165-166): The mean number of sessions was 11.19 (SD = 4.32), with a range of 4 to 20 sessions.

Reviewer #3

1) Can the authors state whether there were any deviations to the pre-registered protocol? if there weren't, it would still be necessary to state that they adhered to this.
All deviations from the preregistration, specifically in regard to upper age limit (p. 16-17, l. 360-362) and revised risk-of-bias tool (RoB 2.0, p. 21, l. 459-461), are referred to in the manuscript.
2) The authors may wish to consider updating their search as it has been six months since

the last search was conducted. It would be necessary to ensure that no new studies are missed during this time.
see response to editor comment #3
3) Please provide key examples of search terms used in the manuscript, rather than providing the whole string in the supplementary materials.
We had omitted them for the sake of brevity but are happy to elaborate in the section “Search strategy”, describing how we set up the search strings and including example terms (p. 16, l. 345-354): We built a search string by combining the concepts “transdiagnostic”, “CBT”, “emotional disorder”, and “RCT” using the AND boolean operator. Each concept included multiple terms connected with the OR boolean operator. The concept “CBT” also covered terms describing the treatment setting (e.g., “Internet-Based Intervention”). We searched for relevant Medical Subject Headings (MeSH), used by the United States National Library of Medicine to index articles in PubMed and MEDLINE (e.g., “Cognitive Behavioral Therapy”, “Anxiety”, “Depression”). In addition, we included terms commonly used in the relevant literature (e.g., “unified”). The resulting string was then slightly adapted according to the search options of the different databases (see Supplementary Table 2 for the complete search strings).
4) Why was the population group limited to adults? Why not include children/adolescents and then conduct sensitivity/subgroup analyses to see if effects vary as a function of age? The authors would need to present some compelling evidence their exclusion
see response to editor comment #2
5) How many authors were contacted for data not reported? This information would be worthwhile to include?
We contacted 35 authors requesting data not reported in their manuscript(s) and sent up to 2 follow-up emails in case of no response. 69% of the authors send us (part of) their data. This information has been added to the manuscript section Study selection and data extraction (p. 18., l. 406-407): We contacted 35 authors and sent up to two follow-up emails in case of no response, 69% of the authors sent us requested data.
6) Was there any rationale for grouping the follow-up time-points in this fashion? Providing some justification would be necessary, otherwise perhaps consider conducting sensitivity analyses ensuring that results do not change due to the cut-off decisions made.
see response to editor comment #2
7) Calculation of pre-post effect sizes is strongly discouraged due the inability to know

which portion of the effect results from naturally occurring changes and which from the intervention (see doi:10.1017/S2045796016000809). The authors are discouraged from reporting these and instead focus on between group effect sizes. This is a possibility given that RCTs were only included.
see response to editor comment #4
8) Please describe the risk of bias indicators in more detail, including what they mean, how they were scored etc. what was the reliability among raters when evaluating risk of bias?
We had kept this brief to streamline the submitted manuscript but are happy to report and discuss the risk of bias indicators in more depth. For transparency, we have also added the codings of the two raters and the reliability calculation to our OSF repository. In the methods section, we added more details on the risk of bias assessment (p. 21, l. 464-472): Each domain is made up of several criteria. For example, the first question in domain (a) asks “Was the allocation sequence random?” which, after consulting the respective manuscript, is answered as “yes”, “probably yes”, “probably no”, “no”, or “no information”. The RoB 2.0 provides examples and decision trees that clearly specify that certain combinations of ratings across questions within a domain result in the risk of bias of that domain being rated as “low”, “some concerns”, or “high”. In domain (a), a “high” risk of bias would be noted if differences between intervention groups were evident at baseline, suggesting a problem with the randomization - regardless of whether no risk of bias was indicated in the evaluation of all other criteria. In the results section, we now report details on inter-rater reliability. Due to the nature of the studies evaluated, there was perfect or near-perfect agreement for a high proportion of risk of bias criteria. For example, unlike in pharmacological studies where a drug or placebo are administered, therapists delivering a psychotherapeutic intervention had to be aware of a participant's assignment to be able to provide the treatment. Thus, raters affirmed that therapists were aware of assignment (“yes” or in one case “probably yes”) and did not use other rating options (e.g., “probably no”, “no”, or “no information”). Given this lack of variability in ratings, we did not calculate Cohen’s kappa, but report the percentage of agreement between raters (p. 9, l. 188-192): Agreement between the two independent raters in coding the risk of bias criteria was strong (M = 90.31%, SD = 7.82%, range = 73.58 - 100%). Instances of disagreement mainly reflect differing levels of a rating, e.g., “yes” vs. “probably yes”, but not the general direction. All ratings and the code for analysis of percentage agreement can be found on the Open Science Framework repository (see sections Data availability and Code availability).
9) Please remove all of the citations on Page 13 and instead place them in a table with the study characteristics. It is too difficult to follow at this point.
We agree and have revised the manuscript accordingly.
10) It would be necessary to provide numbers pertaining to risk of bias indicators. In other words, how many met quality criteria 1, 2...5? What percentage of studies met

all 5 criteria?
We had provided risk of bias assessments of all domains for all studies as well as the percentage of studies showing low/high risk of bias or some concerns in each domain. However, this may not have been clear enough in the text. We have therefore revised the results section on risk of bias assessment and also state that no study was completely free from risk of bias (p. 9-10, l. 192-204): Figure 2 provides an overview of the risk of bias assessment for the five domains rated (see the section Study quality assessment in the Methods) for the individual studies. In Supplementary Figure 1, we also provide a summary plot, depicting the percentage of studies showing low/high risk of bias or some concerns in each domain. We found that, overall, the risk of bias assessment of the majority of included studies showed some concerns and no study was free from any risk of bias. Although there were hardly any concerns regarding bias in the randomization process, all studies showed some concerns regarding blinding of therapists, as they needed to be aware of the protocol they were providing, and assessors, because we only included self-report outcomes. While intention-to-treat (ITT) analyses were conducted in most studies, only few reported comprehensive tests of potential bias in results due to missing outcome data, raising some concerns. Finally, although trial registrations were available for almost all included RCTs, hardly any studies provided an a priori specified analysis plan.
11) Pre-post effect sizes are likely inflated and tell us very little. The authors should consider removing these.
see response to editor comment #4
12) How did the authors detect outliers? What was the criteria for determining an outlier?
We have referenced that studies were excluded if their 95% CI did not overlap with the 95% CI of the overall effect size, in line with a previous meta-analysis of transdiagnostic outcomes (Cuijpers et al., 2023) and as implemented in the "find.outliers" function of the "dmetar" R package. The outlier analyses can also be followed in scripts 7-10 in our OSF repository.

Decision Letter, first revision:

Dear Dr. Meine,

Thank you for your patience as we've prepared the guidelines for final submission of your Nature Human Behaviour manuscript, "A systematic review and meta-analysis of transdiagnostic cognitive behavioral therapies for emotional disorders" (NATHUMBEHAV-23041045A). Please carefully follow the step-by-step instructions provided in the attached file, and add a response in each row of the table to indicate the

changes that you have made. Please also address the additional marked-up edits we have proposed within the reporting summary. Ensuring that each point is addressed will help to ensure that your revised manuscript can be swiftly handed over to our production team.

We would hope to receive your revised paper, with all of the requested files and forms within two-three weeks. Please get in contact with us if you anticipate delays.

Nature Human Behaviour offers a Transparent Peer Review option for new original research manuscripts submitted after December 1st, 2019. As part of this initiative, we encourage our authors to support increased transparency into the peer review process by agreeing to have the reviewer comments, author rebuttal letters, and editorial decision letters published as a Supplementary item. When you submit your final files please clearly state in your cover letter whether or not you would like to participate in this initiative. Please note that failure to state your preference will result in delays in accepting your manuscript for publication.

In recognition of the time and expertise our reviewers provide to Nature Human Behaviour's editorial process, we would like to formally acknowledge their contribution to the external peer review of your manuscript entitled "A systematic review and meta-analysis of transdiagnostic cognitive behavioral therapies for emotional disorders". For those reviewers who give their assent, we will be publishing their names alongside the published article.

Cover suggestions

We welcome submissions of artwork for consideration for our cover. For more information, please see our https://www.nature.com/documents/Nature_covers_author_guide.pdf target="new"> guide for cover artwork.

ORCID

Non-corresponding authors do not have to link their ORCID but are encouraged to do so. Please note that it will not be possible to add/modify ORCID at proof. Thus, please let your co-authors know that if they wish to have their ORCID added to the paper they must follow the procedure described in the following link prior to acceptance:

Nature Human Behaviour has now transitioned to a unified Rights Collection system which will allow our Author Services team to quickly and easily collect the rights and permissions required to publish your work. Approximately 10 days after your paper is formally accepted, you will receive an email in providing you with a link to complete the grant of rights. If your paper is eligible for Open Access, our Author Services team will also be in touch regarding any additional information that may be required to arrange payment for your article.

Please note that *Nature Human Behaviour* is a Transformative Journal (TJ). Authors may publish their research with us through the traditional subscription access route or make their paper immediately open access through payment of an article-processing charge (APC). Authors will not be required to make a final decision about access to their article until it has been accepted. Find out more about Transformative Journals

[REDACTED]

Best regards,
Alex McKay
Editorial Assistant
Nature Human Behaviour

On behalf of

Samantha Antusch

Samantha Antusch, PhD
Senior Editor
Nature Human Behaviour

Reviewer #1:
None

Reviewer #2:
Remarks to the Author:
The authors have satisfactorily addressed my concerns. I commend them on this work.

Reviewer #3:
Remarks to the Author:
Authors have done a terrific job responding to the initial concerns raised. I have nothing further to add.

Final Decision Letter:

Dear Dr Meine,

We are pleased to inform you that your Article "A systematic review and meta-analysis of transdiagnostic cognitive behavioral therapies for emotional disorders", has now been accepted for publication in Nature Human Behaviour.

Please note that *Nature Human Behaviour* is a Transformative Journal (TJ). Authors may publish their research with us through the traditional subscription access route or make their paper immediately open access through payment of an article-processing charge (APC). Authors will not be required to make a final decision about access to their article until it has been accepted. Find out more about Transformative Journals

We welcome the submission of potential cover material (including a short caption of around 40 words) related to your manuscript; suggestions should be sent to Nature Human Behaviour as electronic files

(the image should be 300 dpi at 210 x 297 mm in either TIFF or JPEG format). Please note that such pictures should be selected more for their aesthetic appeal than for their scientific content, and that colour images work better than black and white or grayscale images. Please do not try to design a cover with the Nature Human Behaviour logo etc., and please do not submit composites of images related to your work. I am sure you will understand that we cannot make any promise as to whether any of your suggestions might be selected for the cover of the journal.

With best regards,

Samantha Antusch

Samantha Antusch, PhD
Senior Editor
Nature Human Behaviour